# ORAI channels are critical for receptor-mediated endocytosis of albumin

Bo Zeng[1,2], Gui-Lan Chen[1,2], Eliana Garcia-Vaz[3], Sunil Bhandari[4], Nikoleta Daskoulidou[1], Lisa M. Berglund[3], Hongni Jiang[1], Thomas Hallett[1], Lu-Ping Zhou[2], Li Huang[2], Zi-Hao Xu[2], Viji Nair[5], Robert G. Nelson[6], Wenjun Ju[5], Matthias Kretzler[5], Stephen L. Atkin [1,7], Maria F. Gomez [3] & Shang-Zhong Xu [1]

Impaired albumin reabsorption by proximal tubular epithelial cells (PTECs) has been highlighted in diabetic nephropathy (DN), but little is known about the underlying molecular mechanisms. Here we find that ORAI1-3, are preferentially expressed in PTECs and down-regulated in patients with DN. Hyperglycemia or blockade of insulin signaling reduces the expression of ORAI1-3. Inhibition of ORAI channels by BTP2 and diethylstilbestrol or silencing of ORAI expression impairs albumin uptake. Transgenic mice expressing a dominant-negative Orai1 mutant (E108Q) increases albuminuria, and in vivo injection of BTP2 exacerbates albuminuria in streptozotocin-induced and Akita diabetic mice. The albumin endocytosis is $Ca^{2+}$-dependent and accompanied by ORAI1 internalization. Amnionless (AMN) associates with ORAIs and forms STIM/ORAI/AMN complexes after $Ca^{2+}$ store depletion. STIM1/ORAI1 colocalizes with clathrin, but not with caveolin, at the apical membrane of PTECs, which determines clathrin-mediated endocytosis. These findings provide insights into the mechanisms of protein reabsorption and potential targets for treating diabetic proteinuria.

[1] Centre for Cardiovascular and Metabolic Research, Hull York Medical School, University of Hull, Hull HU6 7RX, UK. [2] Key Laboratory of Medical Electrophysiology, Ministry of Education, and Institute of Cardiovascular Research, Southwest Medical University, Luzhou 646000, China. [3] Department of Clinical Sciences in Malmö, Lund University Diabetes Centre, Lund University, Malmö, 214 28 Malmö, Sweden. [4] Department of Renal Medicine and Hull York Medical School, Hull Royal Infirmary, Hull and East Yorkshire Hospitals NHS Trust, Hull HU3 2JZ, UK. [5] Department of Internal Medicine & Department of Computational Medicine and Bioinformatics, University of Michigan, Ann Arbor, MI 48109, USA. [6] Chronic Kidney Disease Section, National Institute of Diabetes and Digestive and Kidney Diseases, National Institutes of Health, Phoenix, AZ 85014, USA. [7] Weill Cornell Medical College Qatar, PO Box, 24144 Doha, Qatar. Bo Zeng, Gui-Lan Chen and Eliana Garcia-Vaz contributed equally to this work. Correspondence and requests for materials should be addressed to B.Z. (email: zengbo@swmu.edu.cn) or to S.-Z.X. (email: sam.xu@hyms.ac.uk)

Diabetic nephropathy (DN) is a major cause of end-stage renal disease, which is characterized by albuminuria, glomerulosclerosis and progressive loss of renal function. Up to one-third of patients with diabetes develop DN[1]. Moderately increased albuminuria is the earliest detectable sign of diabetic kidney damage and continuous proteinuria causes tubulointerstitial inflammation, scarring and progressive loss of renal function[2]. Glomerular hyperfiltration and reduced reabsorption by proximal tubules are two determinants for albuminuria. Recently, impaired tubular uptake as the cause of albuminuria in the early stages of DN has been highlighted in the development of albuminuria[3,4]. Therefore, an understanding of the molecular mechanisms of protein reabsorption is important for the development of potential therapies.

ORAI channels have been identified as the molecular fingerprints of $Ca^{2+}$-release activated $Ca^{2+}$ (CRAC) channels, the highly $Ca^{2+}$ selective store–operated channels (SOCs) that can be activated by depletion of endoplasmic reticulum (ER) $Ca^{2+}$ stores[5]. Three isoforms of ORAI channels (ORAI1-3) have been identified and each has an intracellular C- and N-terminus, and a transmembrane region with four domains[6]. Store-operated $Ca^{2+}$ entry (SOCE) through ORAI channels triggered by STIM1 is a major mechanism mediating the signals of many hormones, growth factors, cytokines, and neurotransmitters by acting on G protein-coupled receptors (GPCR) and protein tyrosine kinase (PTK) coupled receptors[7,8]. Loss-of-function mutation of ORAI1 causes deficiency of $Ca^{2+}$ release-activated $Ca^{2+}$ current ($I_{CRAC}$) in T-cells, which results in severe combined immune deficiency syndrome[9]; however, the role of ORAI channels in DN is unknown.

Here we report a mechanism for the progressive disorder of DN. We have investigated the expression and function of ORAI

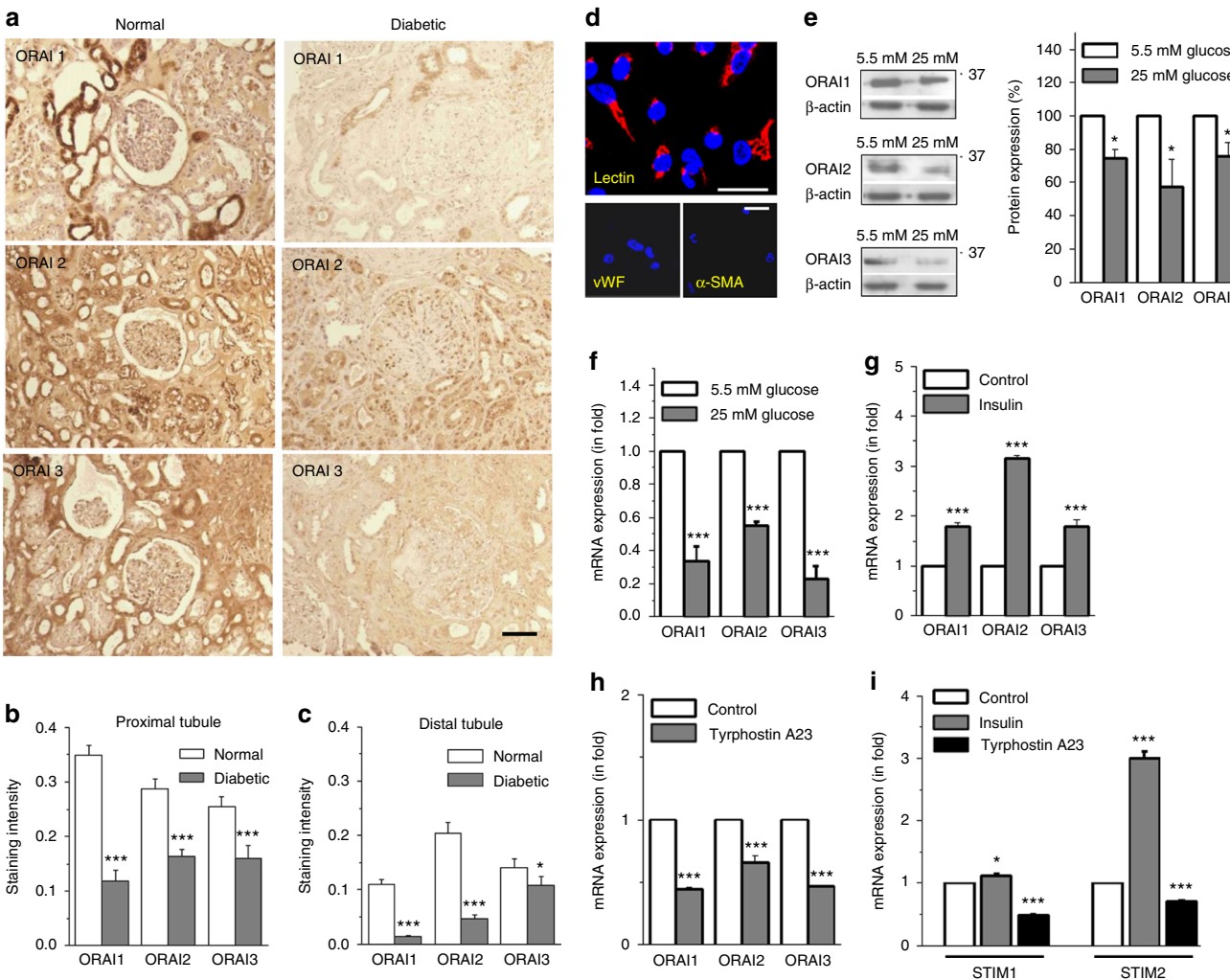

**Fig. 1** ORAIs and STIMs in human kidney and regulation under diabetic condition. **a** Immunostaining for ORAIs in normal and diabetic kidney tissue sections. The diabetic kidney sections showing typical mesangial expansion and accumulation of mesangial matrix material. Scale bar, 100 μm. **b**, **c** Mean ± s.e.m. for the staining intensity (arbitrary unit) in proximal tubules and distal tubules, respectively. The average of nine staining fields for each patient was calculated for proximal or distal tubule staining ($n = 6$ for normal kidney, $n = 8$ for diabetic kidney). Also see staining for STIM1 and STIM2 (Supplementary Fig. 3). **d** Primary cultured human proximal tubular epithelial cells (PTECs) were characterized by lectin staining (red). Scale bars, 50 μm. **e** PTECs were cultured with normal (5.5 mM) and high (25 mM) glucose for 60 h. ORAI proteins were detected by western blotting ($n = 6$). **f** The mRNA of ORAIs was quantified by real-time PCR. The mean data were from 2–3 independent experiments ($n = 6$). **g** The proximal tubular epithelial cells (HK-2) were treated with or without (control) insulin (10 nM) for 48 h and the mRNA was detected by real-time PCR ($n = 6$). **h** HK-2 cells incubated with tyrosine kinase inhibitor tyrphostin A23 (30 μM) for 48 h ($n = 6$). **i** STIM1 and STIM2 expression after insulin (10 nM) and tyrphostin A23 (30 μM) treatment for 48 h ($n = 9$). The β-actin was used as control for relative quantification of mRNA or protein. For PCR experiments, triplicate reactions were set for each gene. The averaged data are displayed as mean ± s.e.m. and the data in **e**–**i** are normalized to control. The data sets are compared by $t$ test. Statistical significance is indicated by $^{*}P < 0.05$, $^{***}P < 0.001$

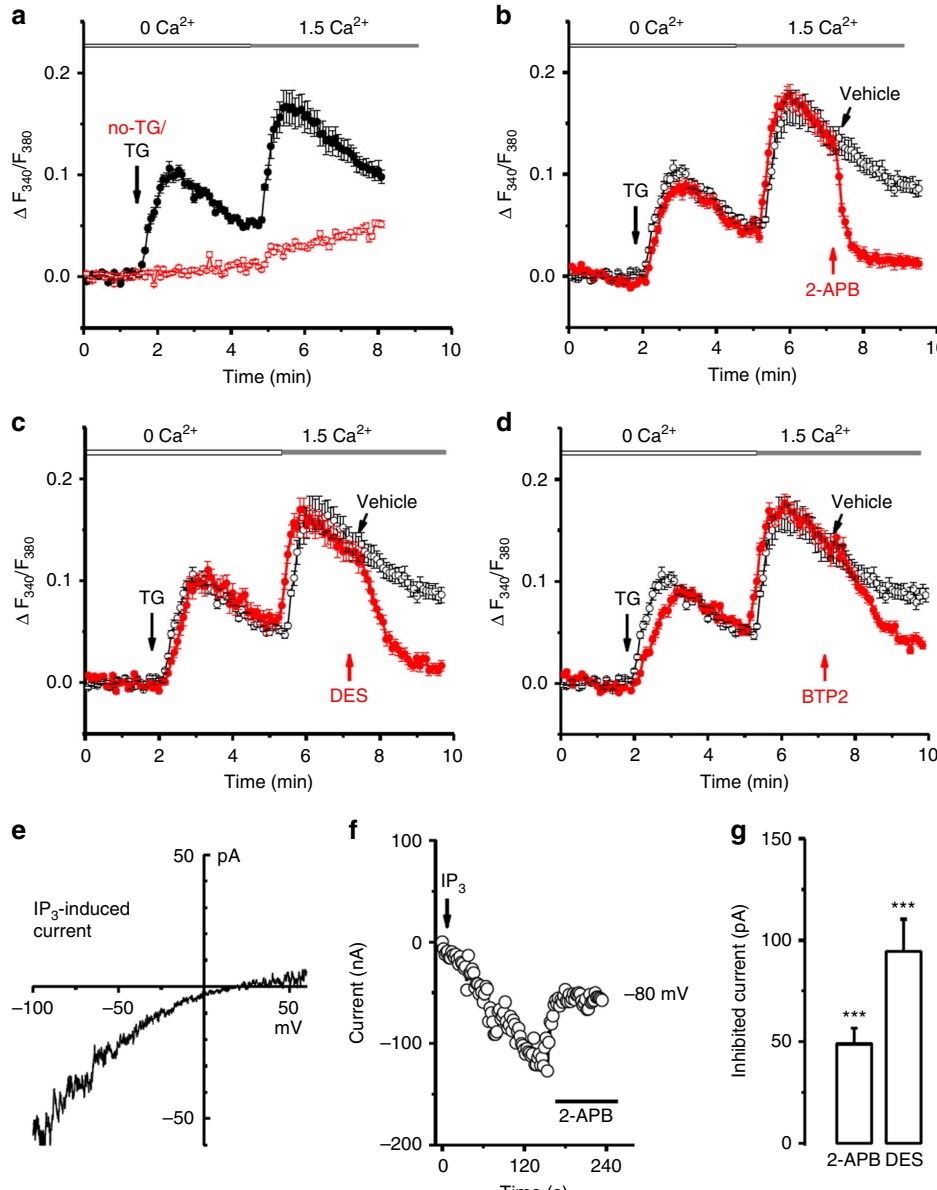

**Fig. 2** Store-operated $Ca^{2+}$ entry and $Ca^{2+}$ release-activated current ($I_{CRAC}$) in proximal tubular epithelial HK-2 cells. **a** Store-operated $Ca^{2+}$ influx induced by thapsigargin (TG, 1 µM, $n = 12$) compared to the group without TG (no-TG, $n = 13$). **b** 2-APB (100 µM, $n = 14$ cells) or vehicle (0.1% dimethyl sulfoxide (DMSO); $n = 13$ cells) was applied after store-depletion by TG and during perfusion with bath solution containing 1.5 mM $Ca^{2+}$. **c** Diethylstilbestrol (DES; 10 µM, $n = 15$ cells) or vehicle (DMSO, $n = 13$ cells) was applied as in (**b**). **d** Perfusion with BTP2 (1 µM, $n = 17$ cells) or vehicle (DMSO, $n = 13$ cells) as in (**b**). **e** Current–voltage (IV) relationship for $I_{CRAC}$ induced by $IP_3$ (30 µM) in the pipette solution. **f** Time course for $I_{CRAC}$ induced by $IP_3$ and the inhibition by 2-APB (100 µM) in HK-2 cells. **g** Mean ± s.e.m. data for $I_{CRAC}$ inhibited by 2-APB (100 µM, $n = 8$) and DES (10 µM, $n = 5$). ***$P < 0.001$. The data sets are compared by $t$ test. Statistical significance is indicated by ***$P < 0.001$

isoforms in proximal tubular epithelial cells (PTECs) using human cell models and biopsies, in vivo diabetic mouse models and transgenic mice. ORAI channels act as key elements in the endocytic process of albumin reabsorption in PTECs. Down-regulation and enhanced internalization of ORAI channel complexes with the endocytic receptors during albuminuria could account for the progressive deterioration of renal function in patients with DN.

## Results

**ORAIs are expressed in human kidney and downregulated in DN.** ORAI1-3 channels were detected in human kidney samples both at the mRNA and protein levels (Supplementary Fig. 1a, b).

They were preferentially localized to kidney tubules, with stronger staining in the proximal tubules than in the distal convoluted tubules (Supplementary Fig. 1c). This is in agreement with rat RNAseq data showing higher tubular than glomerular expression of all ORAI1-3 genes[10] and also with available human RNAseq data (Nephroseq) for ORAI2 expression in human kidney samples from healthy living donors, confirming higher tubular than glomerular expression of this gene (Supplementary Fig. 2a). ORAI1-3 immunostaining in both proximal tubules and distal tubules was weaker in kidney tissue sections from type 1 diabetic patients with DN when compared to non-diabetic controls (Fig. 1a–c; Supplementary Table 1 for patient characteristics). Correspondingly, human RNAseq data from DN patients with estimated glomerular filtration rate (eGFR) ranging between 12

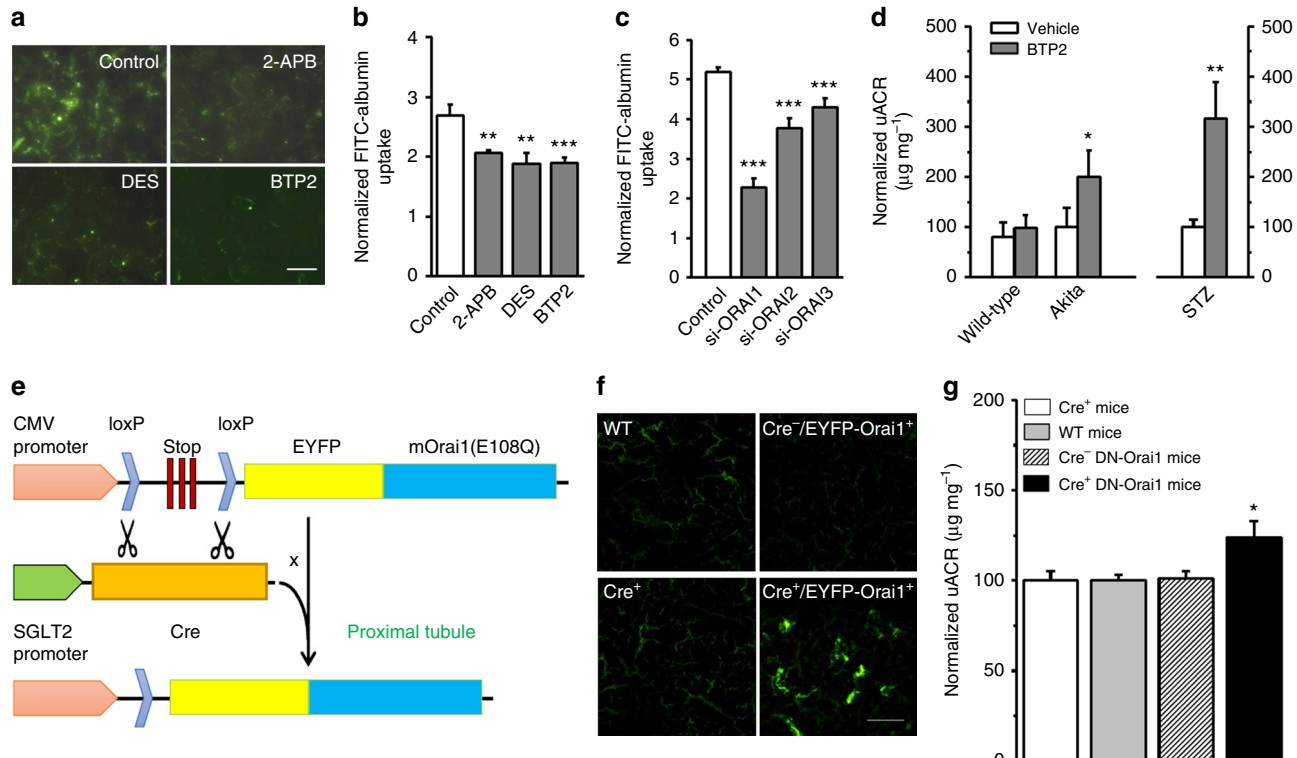

**Fig. 3** ORAI channels determine albumin reabsorption in vitro and in vivo. **a** FITC-albumin uptake by human proximal tubular epithelial HK-2 cells after incubation with FITC-albumin (50 µg mL$^{-1}$) for 60 min with or without 2-APB (100 µM) or DES (10 µM) or BTP2 (1 µM). Scale bar, 25 µm. **b** Mean ± s.e.m. data for the fluorescence of FITC-albumin uptake by PTECs in **a**. The fluorescence was normalized by the total lysed protein (n = 6 for each group). **c** HK-2 cells were transfected with ORAI siRNAs (si-ORAI1/2/3) and scramble siRNA (control) (n = 7 each group). **d** Male Akita type 1 diabetic mice (C57BL/6-Ins2Akita/J) and STZ-induced diabetic male C57BL/6 J mice were injected either with BTP2 (0.29 mg kg$^{-1}$, i.p., b.i.d) for 3 days, or vehicle (saline) as control (n = 9–10 for each group). Age-matched non-diabetic wide-type male mice were also used (n = 13–14). Urine albumin-creatinine ratio (uACR) was measured on day 5. **e** Schematic representation of Cre-LoxP-mediated recombination and generation of a linear product contain one lox-P site and the target gene (EYFP-mOrai1$^{E108Q}$ mutant) via crossing the Cre$^{+}$ mice with proximal tubule-specific expression driven by sodium-glucose cotransporter 2 (SGLT2) promoter. **f** Expression of EYFP-mOrai1$^{E108Q}$ mutant in tubules (Scale bar, 200 µm). **g** Urine albumin-creatinine ratio (uACR) was measured in male DN-Orai1 transgenic mice and littermates on day 32–38 after birth (n = 27, 34, 17 and 17 for Cre$^{+}$, wild-type, Cre$^{-}$-DN-Orai1 and Cre$^{+}$-DN-Orai1 mice, respectively). Average data are presented as mean ± s.e.m. Data sets are compared by t test in (**d**) and by ANOVA in (**b**, **c**, and **g**). Statistical significance is indicated by $^{*}P < 0.05$, $^{**}P < 0.01$, $^{***}P < 0.001$

and 60 showed that expression of ORAI2 in the tubulointerstitium was lower in patients with DN than that in controls, and positively correlated to eGFR (Supplementary Fig. 2b, c), suggesting the expression of ORAI2 in tubules is related to the severity of DN.

Rodents are in general quite resilient to develop diabetic complications as seen in humans and often replicate only early features of DN. We examined ORAI1-3 and STIM1-2 mRNA expression in whole kidney homogenates from Akita mice and STZ-diabetic mice and found increased expression of all targets in diabetic mice (Supplementary Fig. 2d, e). In the Pima Indian study cohort, patients with DN at early stage (normal GFR or hyperfiltration stage), a small increasing trend but not statistically significant was observed for ORAI2 expression in the tubulointerstitium (Supplementary Fig. 2f), suggesting that downregulation of ORAIs is associated with fast kidney function decline during the late stage of DN, but not the early stage of DN. Taken together, the rodent expression and human RNAseq data demonstrate plasticity of ORAI and STIM genes during the development of DN.

To investigate whether changes in ORAI expression are driven by glucose or insulin, in vitro experiments were performed using primary cultured human PTECs. Cells were characterized by positive lectin staining in their apical membrane (Fig. 1d). Both

mRNA and protein levels of ORAI1-3 were significantly down-regulated by high glucose (Fig. 1e, f). Incubation of PTECs with insulin increased mRNA levels of ORAI1-3, while incubation with tyrphostin A23, a protein tyrosine kinase inhibitor that prevents insulin receptor activation, reduced the expression (Fig. 1g, h). Insulin also increased the expression of STIM1 and STIM2, while tyrphostin A23 decreased them (Fig. 1i). STIM1 and STIM2 were also detected in both glomerulus and tubulointerstitium of human kidney (Supplementary Fig. 3a–d). In the Nephroseq data set, STIM1 mRNA intensity was higher in glomeruli and positively correlated to eGFR (Supplementary Fig. 3c, d). Downregulation of STIM1 was also observed in kidney sections from STZ-induced diabetic mice after long-term hyperglycemia (Supplementary Fig. 3e, f). These data demonstrate the expression of ORAIs and STIMs in kidney tubules and downregulation under diabetic conditions.

**Identification of store-operated Ca$^{2+}$ influx in PTECs.** ORAI and STIM constitute SOCs[7], we therefore explored the existence of store-operated Ca$^{2+}$ entry (SOCE) and Ca$^{2+}$ release-activated Ca$^{2+}$ current ($I_{CRAC}$) in PTECs. Store depletion by thapsigargin (TG) resulted in evident SOCE upon restoration of extracellular Ca$^{2+}$, which was significantly inhibited by the SOC blockers,

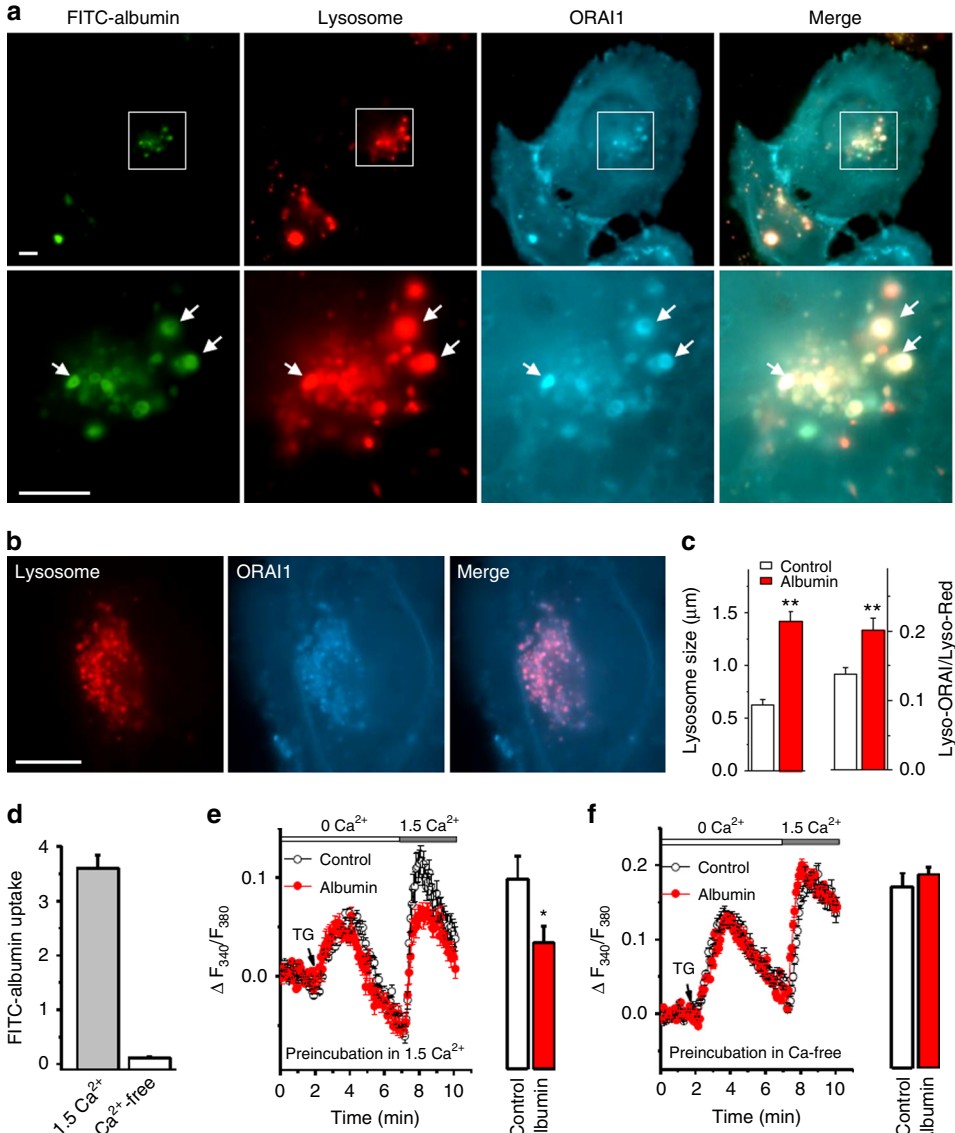

**Fig. 4** Albumin endocytosis enhances ORAI1 channel internalization and impairs SOCE. **a** TIRF microscopy images showing human proximal tubular epithelial cells (PTECs) after incubation with FITC-albumin (50 μg ml$^{-1}$) for 4 h. PTECs were transfected with ORAI1-tagged with CFP. Lysosomes were stained using LysoBrite Red. **b** The PTECs without FITC-albumin treatment. Scale bars, 10 μm for (**a**) and (**b**). **c** Mean lysosome size and fluorescence intensity (ratio of CFP-ORAI1 in lysosomes (Lyso-ORAI) to LysoBrite Red (Lyso-Red) ($n = 32$–50). **d** FITC-albumin uptake in PTECs in 1.5 mM Ca$^{2+}$ bath solution ($n = 30$) or in Ca$^{2+}$-free solution with 0.4 mM EGTA ($n = 30$). **e** Pre-incubation with FITC-albumin for 30 min in 1.5 mM Ca$^{2+}$ solution reduces store-operated Ca$^{2+}$ influx in PTECs. Bar chart shows the mean data for the peak of SOCE in the FITC-albumin incubated ($n = 12$) and control PTECs ($n = 16$). **f** Unaffected SOCE when PTECs were pre-incubated with ($n = 13$) or without ($n = 15$) FITC- albumin for 30 min in Ca$^{2+}$-free solution. Average data are presented as mean ± s.e.m. The data sets are compared by $t$ test. Statistical significance is indicated by $^{*}P < 0.05$, $^{**}P < 0.01$

2-aminoethoxydiphenyl borate (2-APB), diethylstilbestrol (DES) and BTP2 (Fig. 2a–d). The inhibitory effects on SOCE were also observed when cells were pre-incubated with DES, BTP2 and 2-APB prior to the stimulation with TG (Supplementary Fig. 4). In addition, a store-operated current was recorded in PTECs using the same conditions for $I_{CRAC}$ recording in Jurkat cells. The current was gradually evoked by inositol trisphosphate (IP$_3$) in the pipette solution or by TG in both bath and pipette solutions. Current–voltage ($IV$) relationship of IP$_3$-induced current in the PTECs showed an inward rectification ranged from −100 mV to + 50 mV (Fig. 2e, f), which was similar to the current recorded in Jurkat cells (Supplementary Fig. 5). Both 2-APB and DES inhibited IP$_3$- or TG-induced store-operated currents in the PTECs (Fig. 2g). These findings reveal the existence of store-operated Ca$^{2+}$ influx and current in PTECs.

**Inhibition of ORAI impairs albumin reabsorption**. 2-APB has been shown to inhibit ORAI1 and ORAI2, but stimulate ORAI3[11]; however, it is unclear for BTP2 and DES. Using stably transfected STIM1/ORAI1-3 cells, we found that both BTP2 and DES acted as pan inhibitors of the three ORAI isoforms (Supplementary Fig. 6). We next examined FITC-conjugated albumin (FITC-albumin) reabsorption in PTECs using channel blockers and siRNAs. PTECs showed evident intracellular accumulation of FITC-albumin (Fig. 3a). Blockade of SOC activity by 2-APB, DES, or BTP2 significantly reduced FITC-albumin uptake (Fig. 3a, b). Cells transfected with ORAI siRNAs decreased SOCE and reduced the FITC-albumin uptake compared with corresponding scramble siRNA-transfected groups (Fig. 3c, Supplementary Fig. 7). Considering that the silencing efficiencies were 45% for ORAI1, 82% for ORAI2 and 74% for ORAI3 when compared to scramble controls, we could

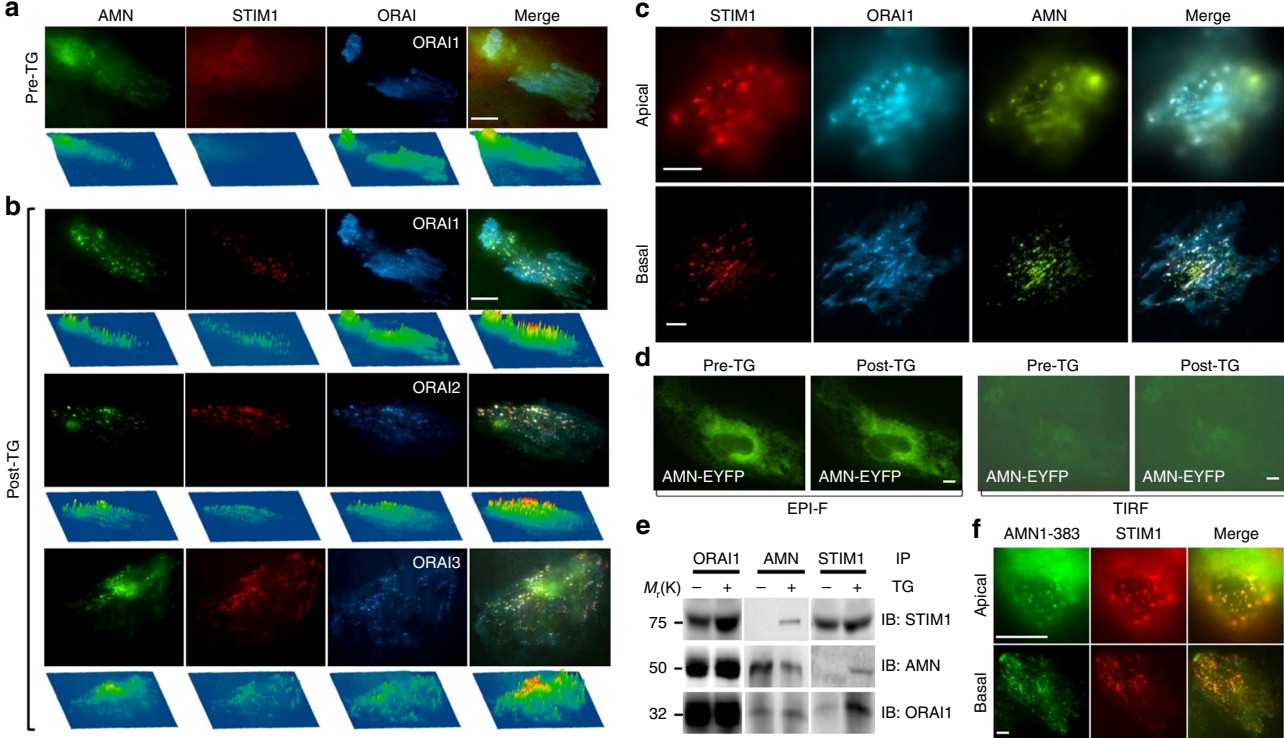

**Fig. 5** Subcellular localization and interaction of amnionless (AMN), STIM1 and ORAIs in PTECs. PTECs were cotransfected with AMN-EYFP, STIM1-mCherry and CFP-ORAIs and fluorescence was examined before and after treatment with thapsigargin (TG, 1 μM) for 10 min using TIRF microscopy with objective ( × 100). **a** Before TG treatment (Pre-TG). **b** After store-depletion with TG (post-TG). The fluorescence distribution pattern and intensity (below each image) are shown for cells transfected with CFP-ORAI1, CFP-ORAI2, and CFP-ORAI3, respectively, plus AMN-EYFP and STMI1-mCherry. **c** Pictures show the localization of STIM1-mCherry, CFP-ORAI1 and AMN-EYFP at the apical and basal membrane of human PTECs after treatment with 1 μM TG for 10 min. **d** No significant clusters were formed before or after treatment with SERCA blocker TG (1 μM) in PTECs transfected with AMN-EYFP alone. Images were taken from Epi-Fluorescence (EPI-F) and TIRF microscopy. **e** Co-immunoprecipitation of STIM1, ORAI1 and AMN endogenously expressed in PTECs with/without TG treatment. **f** Colocalization of intracellular C terminus-truncated AMN (AMN1-183) with STIM1 in PTECs after Ca²⁺ store depletion. Scale bars, 10 μm

conclude that ORAI1 had a greater impact on FITC-albumin uptake than ORAI2 and ORAI3. These in vitro results demonstrate that downregulation of ORAI expression or inhibition of ORAI channel activity impairs albumin uptake in PTECs.

The association of ORAI with proteinuria was also tested in vivo using ORAI channel pan inhibitor BTP2. The genetically modified type 1 diabetic Akita mice (C57BL/6-Ins2$^{Akita}$/J) had no significant albuminuria after 15 weeks of hyperglycemia; however, BTP2 significantly evoked albuminuria in Akita mice. Injection of BTP2 also significantly increased albuminuria in streptozotocin-induced diabetic mice (Fig. 3d). These data provide pharmacological evidence that inhibition of ORAI channel activity in vivo aggravates proteinuria in diabetes.

To further examine the contribution of ORAI1 to albuminuria, transgenic mice with proximal tubule-specific expression of a dominant-negative mutant (DN-Orai1$^{E108Q}$) was generated using Cre-LoxP system (Fig. 3e, f, Supplementary Fig. 8). The mouse Orai1 mutant at E108Q is corresponding to the site of E106 in human ORAI1[12]. SOCE was robustly abrogated in the PTECs transfected with Orai1$^{E108Q}$ mutant (Supplementary Fig. 9b). DN-Orai1 transgenic mice showed evident expression of EYFP-Orai1 mutant in the tubules (Fig. 3f). There was no difference in the ratio of kidney/body weight (Supplementary Fig. 9c) and glycosuria (Supplementary Fig. 10) between the DN-Orai1 transgenic mice and wild-type mice; however, the urinary albumin/creatinine ratio (uACR) was significantly higher in DN-Orai1 transgenic mice (Fig. 3g), suggesting that the loss-of-function mutation of ORAI1 leads to the decreased protein reabsorption in proximal tubules and subsequently proteinuria.

**Ca²⁺-dependent endocytosis of ORAI1 and albumin.** We investigated the mechanism of ORAI channels in albumin endocytosis. FITC-albumin was highly accumulated in the lysosomes of PTECs after incubation for 4 h (Fig. 4a). Interestingly, the CFP-tagged ORAI1 was also endocytosed and co-localized with FITC-albumin in the lysosomes (Fig. 4a). The endocytosis of ORAI1 was further confirmed by lysosome protein assay in normal PTECs without transfection (Supplementary Fig. 11). CFP-ORAI1 was also present in the lysosomes when cells were maintained in medium without FITC-albumin, but the mean size of lysosomes was much smaller and the amount of ORAI1 in lysosomes was significantly less than that in cells incubated with FITC-albumin (Fig. 4b, c), suggesting that ORAI1 channel internalization is enhanced by albumin endocytosis. Extracellular Ca²⁺ is required for albumin uptake because there was almost no FITC-albumin uptake into the cells in Ca²⁺-free solution (Fig. 4d). Incubation with albumin for 30 min in Ca²⁺ solution significantly reduced store-operated Ca²⁺ influx in the serum-starved PTECs, but this reduction was not observed if extracellular Ca²⁺ was omitted (Fig. 4e, f). These results demonstrate that Ca²⁺ is essential for albumin uptake and enhanced ORAI channel internalization may associate with the endocytic receptors.

**ORAI channels physically associate with amnionless.** Amnionless (AMN) is a key protein for receptor-mediated endocytosis[13]. We found that AMN was associated with STIM1 and ORAI channels and formed STIM1/ORAI/AMN clusters at the apical and basal plasma membrane (PM) after store depletion

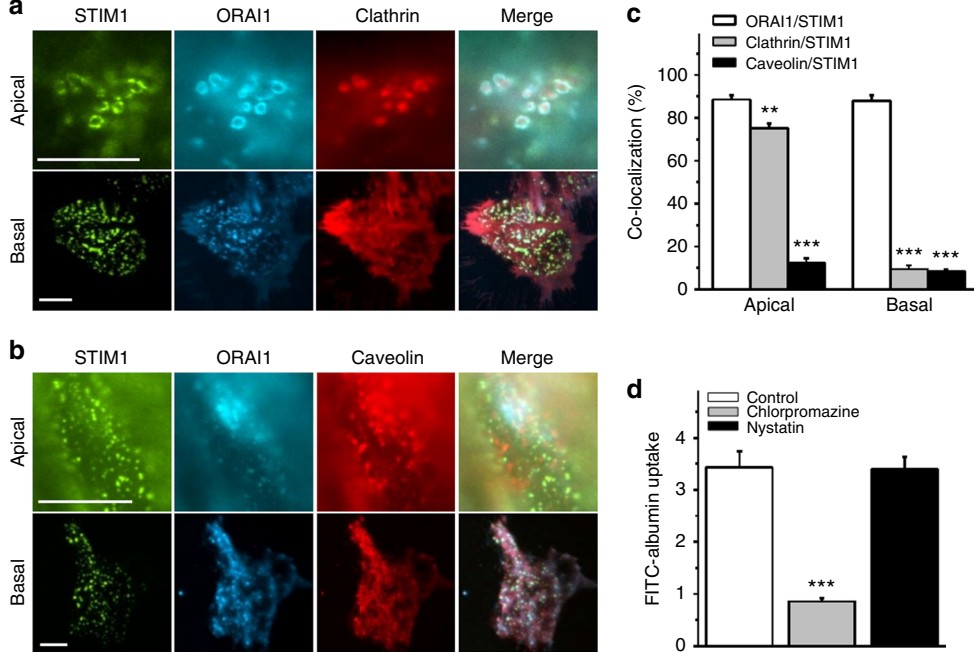

**Fig. 6** Clathrin-mediated endocytosis of albumin and ORAI/STIM complexes upon $Ca^{2+}$ store depletion. **a**, **b**, Localization of STIM1-EYFP, CFP-ORAI1, clathrin-mCherry and caveolin-1-mCherry at the apical and basal membrane of PTECs. **c** Percentage of STIM1 punctum areas that overlapped with ORAI1, clathrin or caveolin-1 ($n = 5$). **d** Effects of endocytosis blockers chlorpromazine (50 μM) and nystatin (50 μM) on the uptake of FITC-albumin by PTECs ($n = 6$ for each group). Average data are presented as mean ± s.e.m. The data sets are compared by ANOVA. Statistical significance is indicated by **$P < 0.01$, ***$P < 0.001$

in the PTECs co-expressing AMN-EYFP, STIM1-mCherry and CFP-ORAIs (Fig. 5a–c). However, transfection with AMN-EYFP alone was unable to form clusters after $Ca^{2+}$ store-depletion (Fig. 5d), suggesting that the movement of AMN requires the presence of STIM1 and ORAIs at a certain level. Co-immunoprecipitation of endogenously expressed STIM1, ORAI1 and AMN showed that ORAI1, but not STIM1, directly interacted with AMN, and such ORAI1-AMN interaction existed in the PTECs with either replete or depleted $Ca^{2+}$ stores (Fig. 5e). The truncated AMN protein (AMN1-383) without cytoplasmic domain was still able to form clusters with STIM1/ORAI1 after $Ca^{2+}$ store-depletion (Fig. 5f), suggesting that AMN is physically associated with ORAI1 through its extracellular domain, and indirectly operated by STIM1 upon $Ca^{2+}$ store depletion.

In human and mouse kidneys under normal condition, ORAI1 was distributed through the apical and basolateral membrane of kidney tubules. After perfusion with TG, ORAI1 was enriched to the brush border of proximal tubules, and colocalized with cubilin, which constitutes the endocytic receptor with AMN and megalin (Supplementary Fig. 12).

**Endocytosis of STIM1-ORAI1 is mediated by clathrin pathway.** Clathrin and caveolin are two key mediators for protein reabsorption, therefore, we examined their subcellular colocalization with STIM1/ORAI1 after $Ca^{2+}$ store depletion in PTECs. Clathrin clearly co-localized with the STIM1/ORAI1 complex at the apical membrane, but not at the basal membrane (Fig. 6a). However, there was no significant colocalization between caveolin and STIM1/ORAI1 complex, neither at the apical nor at the basal membranes (Fig. 6b, c). Moreover, FITC-albumin uptake was inhibited by the clathrin-mediated endocytosis blocker chlorpromazine[14], but not by nystatin that can selectively inhibit lipid raft/caveolae-mediated endocytosis[15] (Fig. 6d). The

internalization of ORAI1 protein into lysosome was also reduced by chlorpromazine (Supplementary Fig. 13). These data suggest that STIM1/ORAI1 is specifically linked to clathrin-mediated endocytosis.

**Spatial organization of F-actin and STIM–ORAI–AMN complexes.** Cytoskeleton reorganization is an alternative mechanism for regulating SOCE and endocytosis[16,17]. We have previously demonstrated that increased F-actin content reduces SOCE, while depolymerization of F-actin restores SOCE[16]. Here we found that although the endoplasmic reticulum (ER) $Ca^{2+}$ store depletion induced STIM1/ORAI1/AMN clustering at the PM–ER junctions, it did not change the organization of F-actin (Fig. 7a). The STIM1/ORAI1/AMN clusters were located between but did not overlap with F-actin bundles at both the apical and basal membranes (Fig. 7b). Disruption of F-actin by cytochalasin D did not affect SOCE and FITC-albumin uptake in PTECs, while calyculin A and U73122, two compounds that increase cortical F-actin, significantly reduced SOCE (Fig. 7c) and FITC-albumin endocytosis (Fig. 7d). These data suggest that F-actin can act as a physical barrier in the PM–ER junctional areas and thus affect SOCE and subsequently protein reabsorption.

## Discussion

Our results indicate that ORAI isoforms are highly expressed in proximal tubules and downregulated in patients with DN. Inhibition of ORAI expression or channel activity by pharmacological blockers or siRNAs or dominant-negative transgenic mice (DN-Orai1) reduces the uptake of albumin by PTECs. In vivo administration of an ORAI channel blocker aggravates proteinuria in diabetic mice. We have also explored the mechanisms of albuminuria caused by ORAI channel inhibition. The enhanced ORAI channel internalization and physical interaction with the

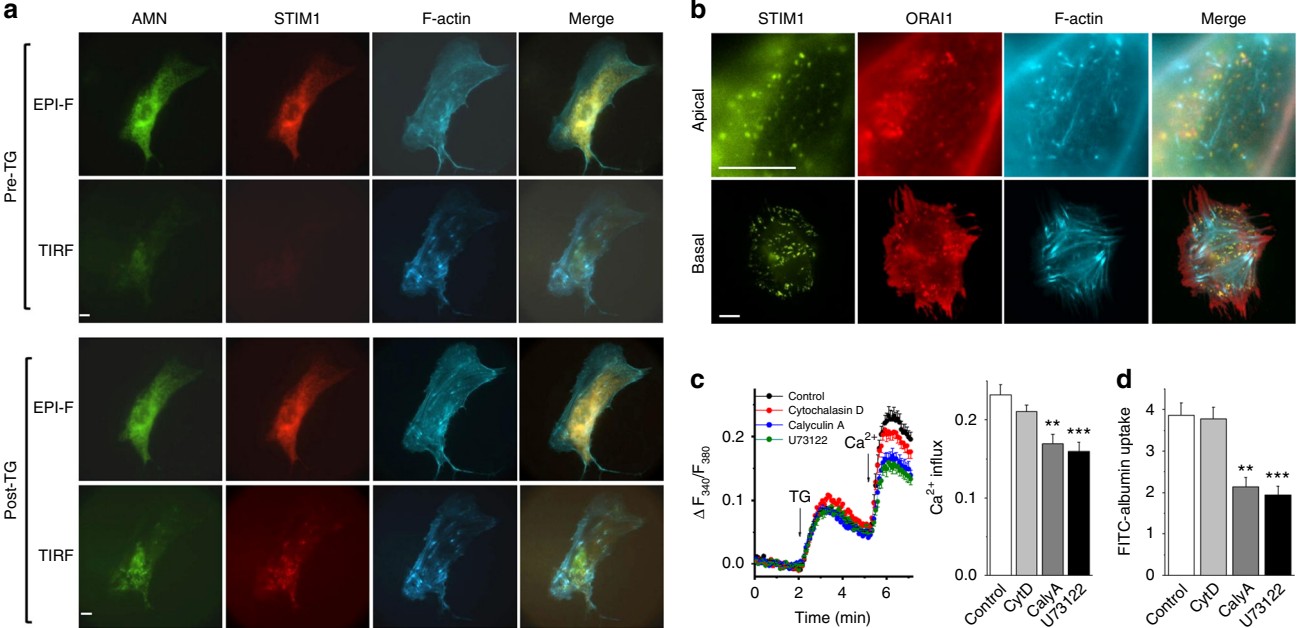

**Fig. 7** Subcellular localization of AMN, STIM1, and F-actin and the role of F-actin in albumin uptake and SOCE. **a** Images taken using Epi-Fluorescence (EPI-F) and TIRF before (pre-TG) and after 1 μM TG treatment for 10 min (post-TG). AMN tagged with EYFP (AMN-EYFP, green) and STIM1 tagged with mCherry (STIM1-Cherry, red). Subplasmalemmal distribution of F-actin was examined by transfecting PTECs with Lifeact-CFP to label F-actin. **b** Localization of STIM1-EYFP, mCherry-ORAI1 and F-actin at the apical and basal membranes after $Ca^{2+}$ store depletion. Scale bars, 10 μm. **c** Influence of cytochalasin D (CytD, 10 μM to depolymerize F-actin, $n = 19$), calyculin A (CalyA, 10 nM, to increase cortical F-actin content, $n = 14$) and U73122 (10 μM, to increase F-actin, $n = 13$) on store-operated $Ca^{2+}$ influx ($n = 12$ in control group). Cells were pre-incubated with each drug for 30 min. **d** FITC-albumin uptake by PTECs was measured after pre-incubation with each drug for 30 min ($n = 6$ for each group). Average data are presented as mean ± s.e.m. The data sets are compared by ANOVA. Statistical significance is indicated by **$P < 0.01$, ***$P < 0.001$

endocytic receptor protein AMN are critical steps for albumin endocytosis, which may account for the reduced capacity of albumin uptake in the diabetic kidney. Our findings suggest that ORAI channel activity is critical for protein reabsorption in the kidney and give insights into ORAI/STIM in the regulation of endocytosis of albumin mediated by the AMN/cubilin/megalin endocytic receptor (Fig. 8). The concept of $Ca^{2+}$ and ORAI-dependent endocytosis could extend to other cell types, since these receptors and channels are expressed in various tissues and are responsible for the reabsorption of other proteins or molecules[18,19]. The cycling of ion channel complexes through clathrin-coated vesicles has important implications in the pathophysiology of diabetic kidney disease and may also be related to the disease processes of ORAI1/STIM1 mutations[9,20–26].

Receptor-mediated endocytosis in proximal tubules is responsible for the reabsorption of albumin in kidney. The endocytic receptor is known to consist of three components: megalin, cubilin and AMN. Megalin (600 kDa) and cubilin (460 kDa) are two large receptors present in apical membranes of absorptive epithelia[18]. Different from megalin, cubilin does not contain a transmembrane segment and its surface expression relies on anchoring to AMN[13], a small (48 kDa) single-transmembrane protein. Cubilin and AMN can form a complex independent of megalin, which is called CUBAM and is responsible for the absorption of cobalamin (vitamin B$_{12}$)[19]. However, for albumin uptake in the kidney, megalin, cubilin and AMN are all required since deficiency of any of the three proteins has been shown to cause albuminuria in patients[27–29]. We have found in this study that ORAI channels directly associate with at least one component of the endocytic receptor in PTECs. Upon $Ca^{2+}$ store depletion, these channels, membrane receptors and intracellular endocytic mediators are concentrated at the PM–ER junctions, suggesting that the contact sites of this specialized membrane are

bona fide hotspots for endocytosis. The STIM1/ORAI1 complex is colocalized with clathrin only at the apical membrane after $Ca^{2+}$ store depletion, which is accordant with the fact that the albumin uptake by PTECs is mainly through the apical surface facing the tubular lumen. This also suggests that the cultured PTECs retain the membrane polarity of epithelial cells, and well mimic the endocytosis mediated by apically expressed cubilin and megalin in vivo[30]. We also excluded the involvement of caveolae/raft-dependent endocytosis in albumin uptake by PTECs, which is consistent with the previous observation in opossum kidney epithelial cells that megalin/cubilin-mediated endocytosis is dependent on clathrin, but not on caveolin[31]. The recruitment of clathrin to STIM1/ORAI/AMN complex may be operated by AMN and megalin, as both of them contain NPXY motifs at their cytoplasmic tails, which function as binding sites for proteins involved in clathrin-coated endocytosis[32]. The consequence of STIM1/ORAI/AMN complex endocytosis may further decrease the SOCE and thus lead to impairment of albumin uptake and progressive albuminuria.

The PM–ER junctional area and F-actin are visualised using TIRF objective after $Ca^{2+}$ store depletion. We found that the endocytosis of albumin in PTECs does not require the structural basis of F-actin, which is in agreement with an early study that cytochalasin D only reduced less than 5% of albumin uptake in opossum kidney epithelial cells[33]. We also showed that excessive F-actin has a negative impact on SOCE and albumin uptake, which is consistent with previous observations by us[16] and others[34] that excessive F-actin could act as a physical barrier between PM and ER, and thus block the interaction between STIM1 and ORAI channels and SOCE. The impairment of SOCE further leads to the reduction of albumin uptake in PTECs seen in the study.

Store-operated $Ca^{2+}$ influx is a predominant $Ca^{2+}$ entry mechanism in non-excitable cells that serves essential functions

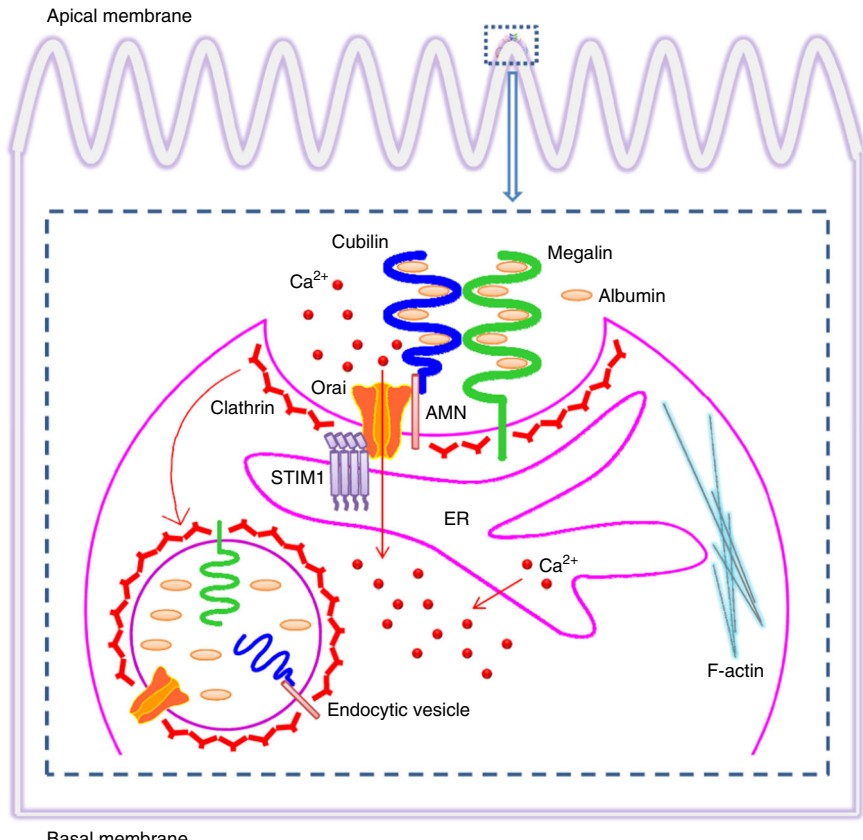

**Fig. 8** Model for ORAI/STIM and endocytic receptors in the process of albumin endocytosis Endoplasmic reticulum $Ca^{2+}$ store-depletion induces intracellular STIM1 movement and subplasmalemmal clustering, and forms ORAI/STIM1 complexes. The ORAI/STIM1 complexes physically associate with amnionless (AMN) to form triple complexes (ORAI/STIM1/AMN). Such triple complexes, together with cubilin and megalin, are endocytosed via clathrin-mediated endocytic mechanism during the PTECs exposure to albumin, which in turns leads to decreased activity of store-operated $Ca^{2+}$ entry in the cells

from gene expression to cell growth[35]. Here we have shown that ORAI channels are preferentially expressed in proximal tubules, but weakly expressed in the glomerulus. The expression was significantly downregulated in in vitro cell models by high glucose and in the biopsy samples from type 1 diabetic patients with DN. Conversely, ORAI isoforms, as well as STIMs, were upregulated by insulin and this upregulation was prevented by an insulin receptor signaling inhibitor. These data suggest that a dynamic balance between glucose and insulin levels is required in determining ORAI channel expression in the tubules. Loss of balance results in the reduction of ORAI expression in patients with DN. Upregulation of ORAIs and STIMs was observed in the cultured vascular cells and tissues from diabetic mice[36] and cultured human mesangial cells[37], such discrepancy in gene expression could be due to the difference of cell type and the stage of diabetes, because rodent models of diabetes normally fail to develop advanced kidney phenotypes and end-stage renal disease, and only mimic the early changes of human diabetic kidney[38,39]. We have not directly compared the difference of store-operated $Ca^{2+}$ entry between normal and diabetic patients due to the limited resource of fresh kidney tissue samples from patients with type 1 diabetes; however, the decreased store-operated $Ca^{2+}$ influx has been reported in retinal microvascular smooth muscle cells from diabetic rats, and such reduction can be restored by application of insulin[40]. SOCE was also increased by insulin-like growth factor 1 (IGF-1) stimulation in a mouse myoblast cell line (C2C12)[41]. Taken together, these evidences suggest that insulin not only activates store-operated $Ca^{2+}$ entry via insulin receptors and tyrosine kinase signaling, but also increases ORAI channel expression

as seen in this study, suggesting that stimulation of insulin signaling may become a molecular strategy to prevent the loss of ORAI channel expression and function in patients with DN.

Increased glomerular filtration is another factor related to DN. The increase in blood pressure and flow may lead to functional and structural changes resulting in high glomerular permeability. The glomerular expression of ORAI channels is not evident in the normal human kidney tissue section in this study, and fewer cells are stained in the glomerulus, however, store-operated $Ca^{2+}$ influx and expression of other $Ca^{2+}$-permeable channels, such as TRP, have been demonstrated in primary cultured mesangial cells[42,43] and podocytes[44], suggesting that other $Ca^{2+}$-permeable channels may be predominant in the regulation of glomerular function. ORAI channels are also expressed in the intrarenal arteries (Supplementary Fig. 1d), suggesting that ORAI or SOC may also be involved in the dysfunction of the renal microcirculation in diabetes. In our DN-Orai1 transgenic mice model, the sodium-glucose cotransporter 2 promoter is used for specific proximal tubular expression, therefore the potential contributions by the changes of glomerular filtration barrier or renal microcirculation may also coexist under certain conditions, which needs to be further investigated.

In conclusion, ORAI store-operated channels exist in human proximal tubules. Impairment of ORAI channel expression or channel activity leads to a decreased albumin absorption by proximal tubular cells and results in proteinuria. $Ca^{2+}$ influx through ORAI channels are critical for the endocytosis of albumin via clathrin-coated vesicles, including the ORAI/STIM/AMN/cubilin/megalin endocytic receptor complexes.

## Methods

**Animals**. Akita type 1 diabetic mice (C57BL/6-Ins2$^{Akita}$/J) were obtained from the Jackson laboratories and bred at Lund University. For the streptozotocin (STZ)-induced type 1 diabetic model, adult C57BL/6 J mice were administered STZ (Sigma-Aldrich; 55 mg kg$^{-1}$ body weight, pH 4.5) or citrate buffer (vehicle) by intraperitoneal (i.p.) injections once a day for 3 days. The animal studies were approved by the Malmö/Lund Animal Care and Use Committee, and abided by the Guide for the Care and Use of Laboratory Animals published by the Directive 2010/63/EU of the European Parliament. Diabetic and control mice were treated twice daily with intraperitoneal (i.p.) injections of BTP2 (Tocris Bioscience, UK; 0.29 mg kg$^{-1}$ body weight per day) for 3 days. Treatment with BTP2 was performed after at least 4 weeks of diabetes for STZ-induced mice, and 10 weeks of age for male Akita mice, which had been diabetic for at least 6 weeks according to the onset age of diabetes at 4 weeks old. Urine was collected during a 2-h period on day 5 after the first injection of BTP2. Albumin concentration was determined by ELISA (Albuwell M kit, Exocell Inc., USA) and creatinine using a picric acid-based assay (Creatinine Companion; Exocell Inc., USA) and an enzymatic colorimetric assay (COBAS, Roche; USA) according to the manufacturer´s instructions.

**Generation of transgenic mice**. The plasmid pRP[Exp]-CMV > loxp-stop-loxp:EYFP-mOrai1(E108Q) was constructed and linearized by NotI. The construct pGL2-sglt2−5pr-mut-Cre was obtained from Dr. Michel Tauc[45]and linearized by ScaI. The F0 transgenic mice (C57/BL6) were generated by pronuclear injection of linearized DNA fragments by Cyagen Biosciences (Suzhou, China). Genotyping of F0 mice and offspring were conducted by PCR using the CMV-F and EcoRI-mOrai1-R primers with a size of 2778 bp, and Psglt2-F and Cre-polyA-R primers with a size of 5238 bp (Supplementary Table 2). Mice carrying the mOrai1(E108Q) and Cre transgenes were bred with wild-type C57BL6 mice separately and then crossed to generate double-positive (mOrai1(E108Q)$^{+}$/Cre$^{+}$) offspring. The transgenic mice were bred in Southwestern Medical University and the experimental procedures were approved by the Ethical Committee of Southwest Medical University.

**Cell culture and transfection**. The full length of human ORAI1, ORAI2 and ORAI3 cDNAs were amplified from human aortic endothelial cells (PromoCell, Heidelberg, Germany), and clathrin light chain and caveolin-1 were amplified from human PTECs by RT-PCR and cloned into a tetracycline-regulatory vector[16]. The plasmid cDNAs of ORAIs tagged with cyan (CFP) or red (mCherry) fluorescent proteins were stably expressed in HEK-293 T-REx cells (Invitrogen, UK). For electrophysiological experiments, the tetracycline-inducible ORAI1-3 cells were co-expressed with STIM1-EYFP. The gene expression of ORAIs was induced by 1 µg ml$^{-1}$ tetracycline for 24–72 h before recording. The non-induced cells without addition of tetracycline or the non-transfected cells were used as controls. Cells were grown in DMEM-F12 (Gibco, UK) medium containing 10% fetal calf serum (FCS), 100 units ml$^{-1}$ penicillin and 100 µg ml$^{-1}$ streptomycin. Cells were maintained at 37 °C under 95% air and 5% $CO_2$ and seeded on coverslips prior to experiments.

The human proximal tubular cell line (HK-2) was purchased from LGC standards (Catalog number CRL-2190, UK). These cells were originally derived from normal adult human renal cortex and immortalized by transduction with human papilloma virus 16 (HPV-16) E6/E7 genes. Cells were maintained in DMEM/F-12 medium with 5 mM glucose and supplemented with 10% FCS, 10 mM HEPES and antibiotics. Cells were seeded on coverslips for patch recording or in 35-mm dishes at 80% confluence for albumin uptake assay. Electroporation was performed for siRNA transfection into HK-2 cells using Neon® transfection system (Invitrogen). The concentration of ORAI siRNAs was 200 nM. The efficiency achieved ~90 % at optimized conditions (20-ms pulse with 1300 V amplitude), which was assessed by cotransfection with a red fluorescent protein (DsRed) plasmid cDNA as a reporter. ORAI siRNAs were purchased from Sigma and the sequences were given in Supplementary Table 2. The scramble siRNA was also from Sigma. The successful silencing of ORAI expression by siRNAs was confirmed by real-time PCR, yielding a 45% reduction of expression for ORAI1, 82% for ORAI2 and 74% for ORAI3 when compared to scramble controls. Jurkat cells were purchased from LGC standards (Middlesex, UK) and cultured in RPMI 1640 medium containing 10 % FCS, 100 units ml$^{-1}$ penicillin and 100 µg ml$^{-1}$ streptomycin.

**Primary culture of human proximal tubular cells**. Proximal tubular cells were isolated from human renal tissue after nephrectomy from portions of the kidney not involved in renal cell carcinoma. The kidney tissue was collected in cold Hank's buffered salt solution (Invitrogen) and transported into the lab immediately. The tissue was cut into approximately 2–8 mm³ pieces by using a sharp blade then the fragments were incubated in 35-mm culture dishes using DMEM/F12 medium with Glutamax (Invitrogen) containing 10% FCS, 100 units ml$^{-1}$ penicillin and 100 mg ml$^{-1}$ streptomycin. Cells were kept at 37 °C in the incubator with a humidified atmosphere of 5% $CO_2$ in air. After 24–48 h culture, cells with the same shape were transferred into a T-75 flask to expand the cell numbers. The cultured proximal cells were confirmed by lectin staining. Cells at passage 2−3 were used to avoid age-dependent phenotypic changes. This study was approved by the Hull and East

Riding research ethical committee and all patients gave their informed consent in accordance with the principles of the Declaration of Helsinki.

**Quantitative RT-PCR**. Total RNA was extracted from snap-frozen normal kidney samples using Trizol (Invitrogen). The RNA was reverse transcribed with moloney murine leukemia virus reverse transcriptase using random primers and oligo dT primers (Promega). Quantitative RT-PCR was performed using StepOne™ Real-Time PCR System (Applied Biosystems, UK). The primer set was designed across introns to avoid genomic DNA contamination and synthesized by Sigma-Genosys. The PCR primer sequences were given in Supplementary Table 2. Each reaction volume was 10 µl, which contained 1 × SYBR Green PCR master mix (Applied Biosystems), 5 µl cDNA, 0.75 µl 300 nM forward primer, 0.75 µl 300 nM reverse primer. Human β-actin was used as an internal standard for quantification. Water was used as a non-template control and non-reverse transcribed samples were run in parallel to exclude genomic DNA contamination. The PCR cycle was programmed as an initial cycle of 50 °C for 2 min followed by 95 °C for 10 min, then 50 repeated cycles of 95 °C for 15 s denaturation and 54 °C annealing temperature for 30 s, and primer extension at 72 °C for 30 s. The reactive conditions for real-time PCR were optimized by monitoring the melting temperature curve and by electrophoresis on agarose gels. The PCR products were confirmed by direct sequencing.

**Western blotting and co-immunoprecipitation**. Kidney samples were homogenized. Cells were lysed with 2 × sample buffer and the proteins were separated on 10% SDS-PAGE gel before transferring onto a nitrocellulose membrane[46]. The blot was then incubated with polyclonal anti-ORAI1 (ACC-060), anti-ORAI2 (ACC-061) or anti-STIM1 (ACC-063) at 1:500 dilution (Alomone Labs, Jerusalem, Israel), and anti-ORAI3 (#4117) at 1:500 dilution (ProSci Incorporated, Poway, CA, USA) overnight at 4 °C, respectively, and washed with phosphate buffered saline (PBS) and then incubated with a goat anti-rabbit IgG-HRP (1:2000 dilution) (A0545, Sigma). The specific binding of anti-ORAI1 and anti-ORAI2 antibodies was confirmed by using the lysates from HEK293 cells overexpressing ORAI1 and ORAI2 proteins tagged with mCherry fluorescence (Supplementary Fig. 14). Rabbit anti-β-actin (1:2000 dilution) (SAB4301137, Sigma) was used as an internal standard for protein quantification. Visualisation was carried out using ECL$^{plus}$ (Amersham Biosciences) and developed onto an X-ray film or photographed by a gel document system. The quantification was carried out using ImageJ (NIH, USA) software. Immunoprecipitation of endogenously expressed STIM1, ORAI1 and AMN proteins in PTECs was performed with a Pierce Classic IP Kit (Thermo Fisher Scientific, USA), and rabbit anti-STIM1 at 1:500 (ACC-063, Alomone Labs, Jerusalem, Israel), rabbit anti-Orai1 at 1:500 (sc-68895, Santa Cruz) and mouse anti-AMN at 1:500 (sc-365384, Santa Cruz) antibodies were used.

**Immunohistochemistry**. Kidney biopsies collected from diabetic type 1 patients for clinical reasons and paraffin blocks containing nephrectomy samples from non-diabetic individuals were used with ethical approval by local research ethics committee. The diabetic kidney samples were confirmed by pathological and clinical diagnosis. Samples from 8 patients (6 males and 2 females, average age of 51.3 ± 4.2 years) were used, all presenting histopathological alterations, including mesangial expansion, arteriolar hyalinosis, global glomerular sclerosis, accumulation of mesangial matrix material, and renal interstitial expansion. Six age-matched non-diabetic normal kidney samples from patients with kidney tumor were stained in parallel for comparison as normal controls. Paraffin-embedded kidney tissue sections with a thickness of 3 µm were immunostained with rabbit anti-ORAI1 (ACC-060), anti-ORAI2 (ACC-061), anti-ORAI3 (Catalogue No. 4117), anti-STIM1 (ACC-063) and anti-STIM2 (ACC-064) antibodies using VECTASTAIN ABC kit (Vector Laboratories). ORAI1, ORAI2, ORAI3, STIM1 and STIM2 primary antibodies at 1:250 dilutions were used and the tissue sections were incubated at 4 °C overnight, followed by biotinylated anti-mouse/rabbit immunoglobulins at 1:1000 for 20 min. The peroxidase conjugated lectin from Arachis hypogaea (Sigma) was used as a positive staining control. Incubation with antigen pre-absorbed antibodies or without primary antibody was used as a negative control. Immunostaining was quantified by imaging software (Image-Pro Plus, Media Cybernetics, USA), under blind conditions. The anti-Orai1 (sc-68895), anti-Orai2 (sc-292103) and anti-Orai3 (sc-292104) antibodies purchased from Santa Cruz Biotech (Dallas, USA) were also used at 1:100 dilution for immunostaining to confirm the tissue distribution of ORAIs in human kidney.

**Immunofluorescence**. Frozen kidney tissue sections (10 µm thickness) were fixed with 4% paraformaldehyde and permeabilised with −20 °C methanol for 1 min and 0.1% Triton X-100 in PBS for 2 h at room temperature. Sections were incubated in 1% bovine serum albumin (BSA) and then in the appropriate ORAI primary antibodies at 1:200–500 dilutions in PBS with 1% BSA at 4 °C overnight. After three times wash with PBS, the tissue sections were then incubated in the sheep anti-rabbit IgG conjugated with FITC (1:160; Sigma) for 2 h at room temperature. Double staining was performed for some experiments using Cy3 conjugated monoclonal anti-α-smooth muscle actin (1:200; Sigma) or TRITC-conjugated lectin (Sigma). After wash with PBS, sections were mounted using Vectashield mounting medium containing DAPI (Vector Laboratories). The staining was

photographed using a laser confocal microscope acquisition software. For paraffin-embedded kidney sections (4 μm thickness), primary antibodies at 1:100 dilution, including rabbit anti-Orai1 (ACC-060 or ACC-062 for mouse tissue, Alomone Labs), mouse anti-AMN (MAB1860, R&D Systems Inc), goat anti-cubilin (sc-20607, Santa Cruz) and rabbit anti-megalin (D160443, Sangon, Shangai, China), and secondary antibodies including donkey anti-rabbit (Alexa Fluor 488), anti-mouse (Alexa Fluor 555) and anti-goat (Alexa Fluor 647) IgG (ThermoFisher) were used.

**Total internal reflection fluorescence microscopy.** Total internal reflection fluorescence (TIRF) microscopy was performed using a Nikon CFI Apochromat TIRF objective (×100, 1.49 NA) and sCMOS camera (ORCA-Flash4.0 V2, Hamamatsu, Japan) mounted on an Eclipse Ti-E inverted microscope with Perfect Focus System (PFS; Nikon). Imaging was performed on PTECs expressing STIM1, ORAI1-3, AMN, clathrin light chain and caveolin-1 tagged with cyan (CFP), enhanced yellow (EYFP) or mCherry fluorescent proteins as indicated. CFP, EYFP and mCherry were excited by 405-, 488- and 561-nm lasers with corresponding filter lens, respectively. Colocalization analysis was performed with NIS-Elements AR v4.30 (Nikon) and 3D surface plot of fluorescent intensities was generated by Image-Pro Plus 6.0 (Media Cybernetics). Three to five experiments were performed for each condition.

**Albumin uptake assay.** Experiments with fluorescein isothiocyanate (FITC)-albumin were performed on cells grown on 35-mm culture dishes to allow microscopic examination and protein extraction. HK-2 cells were incubated with serum free DMEM/F-12 medium for 24 h and treated with ORAI channel inhibitors, endocytosis blockers or F-actin regulators for 30 min, then FITC-conjugated albumin (FITC-albumin) (Sigma) was added to achieve a final concentration of 50 μg ml$^{-1}$. After 60 min incubation at 37 °C, the cells were washed with cold serum free medium containing 10 mg ml$^{-1}$ of unlabelled albumin, and washed again with cold PBS six times. The cells were lysed in H$_2$O by two freezing and thawing cycles, followed by sonication. The supernatant was used for fluorescence and protein assays. The intensity of FITC fluorescence was measured by a fluorophotometry with an excitation/emission wavelength at 485/532 nm. The cells were also photographed using a fluorescent microscope.

**Electrophysiology and Ca$^{2+}$ measurements.** Whole-cell patch recordings were performed at room temperature (23–26 °C) as we described previously[36,46]. Briefly, signal was amplified with an Axopatch 200B patch clamp amplifier connected to a digitizer Digidata 1440 and controlled with pClamp software 10 (Molecular Devices). A 1-s ramp voltage protocol from −100 mV to +100 mV was applied at a frequency of 0.2 Hz. Signals were sampled at 10 kHz and filtered at 3 kHz. The buffered pipette solution contained (mM): 145 Cs-methanesulfonate, 10 BAPTA, 10 HEPES, and 8 MgCl$_2$, and the pH = 7.2 adjusted using CsOH. The standard bath solution contained (mM): 120 NaCl, 2.8 KCl, 10 CsCl, 2 MgCl$_2$, 10 CaCl$_2$, 10 HEPES, and 8 D-Glucose. The pH was adjusted to 7.4 with 1 M NaOH. The recording chamber had a volume of 150 μl and was perfused at a rate of about 2 ml min$^{-1}$. Thapsigargin (1 μM) was added in both pipette and bath solution to deplete the endoplasmic reticulum calcium store.

For Ca$^{2+}$ imaging experiments, HK-2 cells were seeded on coverslips for 24 h and incubated with Fura-PE3 AM (1 μM) for 30 min at 37 °C in Ca$^{2+}$ free standard bath solution. The ratio (F$_{340}$/F$_{380}$) of Ca$^{2+}$ dye fluorescence was measured by a Nikon Ti-E system with NIS-Elements software as we reported previously[16,36]. All the experiments were performed at room temperature.

**Lysosome protein assay.** Lysosomes were isolated from HK-2 cells by ultracentrifuge using a Lysosome Enrichment Kit (Thermo Fisher Scientific, USA). Briefly, cells (~1.5 × 10$^8$) were detached from the culture dishes with a cell scraper, washed with ice-cold PBS and spun down at 500 g for 10 min at 4 °C. The cell pellet was suspended in 800 μl Lysosome Enrichment Reagent A and homogenized by passing through a 22-gauge syringe needle 10 times. The lysate was mixed with 800 μl Reagent B and centrifuged at 500 g for 10 min at 4 °C. The supernatant (post-nuclear supernatant, PNS) was mixed with 500 μl OptiPrep$^{TM}$ Cell Separation Media, transferred to the top of the density gradients in an ultracentrifuge tube, and centrifuged at 145,000 × g for 2 h at 4 °C. The lysosome band close to the top of the density gradients was collected and diluted with 3 volumes of PBS, and centrifuged at 18,000 × g for 30 min at 4 °C. The pellet was then lysed for protein analysis. Antibodies in the Organelle Localization IF Antibody Sampler Kit (Cell Signaling Technology, USA) were used to examine the presence of different organelles at each step during lysosome isolation. Biotin-conjugated goat anti-rabbit IgG secondary antibody (AP132B; 1:1000 dilution,) and HRP-conjugated streptavidin (#18-152, 1:5000) from Millipore (Darmstadt, Germany) were used to detect ORAI1 in lysosomal proteins.

**Microarray data set and bioinformatics.** Human kidney tissue (glomeruli or tubuli) was microdissected as described previously[47]. The expression data of genes of interest in microdissected glomeruli or tubulointerstitium were extracted from the Nephroseq (https://www.nephroseq.org) diabetic data sets[48–50] and the expression data from 22 Southwestern American Indians enrolled in a randomized,

placebo-controlled clinical trial to evaluate the renoprotective efficacy of losartan in type 2 diabetes (clinical trial reg. no. NCT00340678, clinicaltrials.gov)[51] with early stage of DN were used. For Nephroseq data sets, the normalized expression data (median center intensity) were log2-transformed and batch-corrected. Pearson correlation coefficient was used to evaluate association between gene expression level and eGFR.

**Statistics.** All values are expressed as mean ± s.e.m. Paired $t$ test was used to assess statistical differences between two groups, and one-way ANOVA with Bonferroni or Dunnett's post hoc analysis was used for multiple group comparisons with significance indicated if $P < 0.05$.

**Data availability.** The data supporting the findings of this study are available within the article and its Supplementary Information Files or from the corresponding authors upon reasonable request.

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

## Acknowledgements

We thank G. Cooksey, S. Krauss and S. Hetherington (Castle Hill Hospital, Hull and East Yorkshire Hospitals NHS Trust) for kindly providing human kidney samples; N. Watson and A. Green for technical help. This work was supported in part by the Hull York Medical School Pump Priming Award, British Heart Foundation and Leverhulme Trust (to S.-Z.X.); National Natural Science Foundation of China (31300965 to G.-L.C. and 31300949 to B.Z.); China Scholarship Council (to B.Z.); University PhD studentship (to N.D.); National Institute of Diabetes and Digestive and Kidney Diseases (NIDDK) (P30 DK081943, to M.K.); Intramural Research Program of the National Institute of Diabetes and Digestive and Kidney Diseases (to R.G.N.); and by the Swedish Research Council (2011-3900 and 2013-0700), Albert Påhlsson and Diabetes foundations, and the Swedish Foundation for Strategic Research (to M.F.G.). This project has received funding from Innovative Medicines Initiative Joint Undertaking under grant agreement [No. 115006 (SUMMIT) and No. 115974 (BEAt-DKD)], comprising funds from the European Union's Seventh Framework Program [FP7/2007-2013] and Horizon 2020 research and innovation programme and EFPIA (to M.F.G., S.-Z.X., M.K.).

## Author contributions

B.Z., G.-L.C. and E.G.-L.: Coordinated the experimental work and data analysis, generated and validated mice, performed experimental studies, and proposed ideas for experiments. S.B.: Provided clinical kidney samples and intellectual input. N.D. performed the real-time PCR experiments. L.M.B.: Contributed the work on diabetic mice. H.J., T.H., L.-P.Z., L.H., and Z.-H.X.: Performed studies on HK-2 cells. V.N., R.G.N., W.J. and M.K.: Provided the human kidney RNA-seq data sets and analyzed the data. S.L.A., M.F.G.: Provided intellectual input and advised on manuscript. All authors commented on the manuscript. B.Z.: Wrote part of the manuscript and generated the transgenic mice. S.-Z.X.: Initiated the project, generated research ideas, led and coordinated the project, interpreted the data and wrote the manuscript.

## Additional information

**Competing interests:** The authors declare that they have no competing financial interests.

