## [Peer Review File · Nature Communications]

Reviewers' comments:

Reviewer #1 (Remarks to the Author):

In the manuscript the authors report that "ORAI channels are critical for receptor-mediated endocytosis of albumin". They found that: 1) ORAI channels were expressed in proximal tubules and were down-regulated in patients with diabetic nephropathy; 2) impaired the expression or function of ORAI channels led to decrease of albumin uptake and resulted in proteinuria; 3) ORAI channels regulated albumin uptake through Ca²⁺-dependent and clathrin-coated vesicle-mediated endocytosis. The overall findings are interesting and the paper is well written. However, I feel that some conclusions of the study require more supporting evidence and/or clarification.

1) The authors suggested that "The data demonstrate the expression of ORAIs and STIMs in kidney tubules and the down-regulation under diabetic conditions". However, Fig.1i showed that insulin pathway might regulate the mRNA levels of STIMs in PTECs, it is better to show the expression and distribution of STIMs in the kidney biospecimen.

2) The authors suggested that "Inhibition of ORAI store-operated channels impairs albumin uptake in PTECs and aggravates diabetic proteinuria in vivo". It is not clear whether ORAI1~3 differentially contribute to the [Ca²⁺]_i elevation induced by TG in PTECs. In Fig3.c, ORAI1 siRNA had better effects than ORAI2 and ORAI3 siRNAs in regulating albumin uptake. The transfection efficacy and known-down efficacy of each siRNAs should be indicated and additional scramble siRNAs as controls are needed.

3) There were no significant differences in uACR between STZ mice or Akita mice and their controls? How about the expression of ORAIs and STIMs in kidney tubules in diabetic mice? And how were the changes of [Ca²⁺]_i elevation induced by TG in PTECs from these mice?

4) The authors showed that ORAIs preferentially expressed in kidney proximal tubules, however, ORAIs and STIMs are not sole expressed in PTECs. Therefore, intraperitoneal injection of BTP2 increased the uACR in diabetic mice which did not support the conclusion that impairs the function of ORAIs in PTECs aggravates diabetic proteinuria in vivo.

5) The authors concluded that "ORAI1 is endocytosed into lysosomes with albumin in Ca²⁺-dependent manner", based on observation that CFP-tagged ORAI1 co-localized with FITC-albumin and lysosomes. They did not present direct evidence showing that ORAI1 has been internalized. It has been reported that red fluorescent protein-tagged ORAI1 forms intracellular artificial puncta that colocalize with lysosomes [1]. Therefore, the authors should carefully address their conclusion by using fluorescent protein as tag here.

6) The authors did not present direct evidence that STIM/ORAI could be internalized through a clathrin-dependent pathway. Results from colocalization of over-expressed proteins are not enough. They also did not clearly explain how [Ca²⁺]_i regulates albumin uptake in PTECs in their studies.

Reference:

1. Han LY*, Zhao YH*, Zhang X, Peng JX, Xu PY and Huan SY. RFP tags for labeling secretory pathway proteins. BBRC, 447 (2014) 508-512.

Reviewer #2 (Remarks to the Author):

Diabetic Nephropathy is a common complication of DM and a frequent cause of end stage renal disease. Failure to endocytose albumin in the proximal renal tubules contributes to the proteinuria

and renal damage seen in DM. In the present studies by Zeng et al, the authors implicate Orai1 channels in the receptor mediated uptake of albumin. This idea is novel and may have implications for potential therapeutic strategies for diabetic kidney disease. Specifically, STIM1 and Orai1 were diminished in the PRT of akita mice, a model of DM. If these studies can be validated in intact mice or humans, this manuscript would have a broad appeal.

-The main idea is that loss of SOCE by siOrai1 or blocking with BTP2 leads to greater albuminuria because of impaired endocytosis of alb. Studies show this is a specific process involving amnionless signaling.

-A series of studies demonstrate the reduced expression in diabetic kidneys (mice). This finding would be strengthened by current measurements of the channel in primary cells. It would be useful to also show that other currents common to these cells are not altered.

-Some of the figures are missed labeled in the text.

-Studies using differential centrifugation and cell fractionation would strengthen the link between Orai1, AMN and STIM1. These studies could be done in the presence and absence of TG (and TG+BTP2) to show if store depletion influenced alb endocytosis. As it stands fluorescent studies are helpful but the isolation of the different fraction would provide quantitative data.

-At present, renal disorders have not been described in patients harboring mutations in STIM1 and Orai1. It would be critical to show that STIM1 or Orai null mice (or gain of function mice) have increased alb in urine.

Reviewer #3 (Remarks to the Author):

The manuscript by Zeng et al. entitled "ORAI channels are critical for receptor-mediated endocytosis of albumin" aims at identifying the molecular mechanism associated with impaired albumin reabsorption in diabetic nephropathy (DN). The authors reveal down-regulation of ORAI1-3 expression in PTECs from patients with DN that can be mimicked by hyperglycaemia or blockade of insulin signaling. Blocking Orai activity either by inhibitors or siRNA impairs albumin uptake. This uptake is Ca²⁺ dependent and leads to ORAI1 internalization, a process involving amnionless association with ORAI1. Moreover, albumin endocytosis is linked via clathrin to STIM/ORAI1. This manuscript is interesting and timely. The experiments appear carefully conducted and most of the conclusions justified. However, I would like the authors to address the following points to strengthen their manuscript

Major points:

- 1) It is shown in Fig. 4d that albumin uptake is dependent on the presence of extracellular Ca²⁺, and it has been suggested that Ca²⁺ influx through ORAI channels is critical for this endocytosis of albumin. To render this mechanism feasible, it would require that ORAI channels are already activated under conditions present in Fig. 4D. Is this indeed the case and when yes what triggers their activation?
- 2) It has been shown in Fig. 3c that silencing ORAI1-3 reduces albumin uptake in HK-2 cells. It should be shown here as well that preferentially ORAI1 knockdown leads to a reduction of Ca²⁺ influx, both in Ca²⁺ fluorescence and electrophysiological measurements.
- 3) I'm a little puzzled by the I/V relationship and time-course presented for Ca²⁺ currents in Fig. 2e, f: (i) Some current is already active at the beginning, particularly at +80mV, (ii) I would have expected a much faster activation with IP₃ as it actively depletes ER stores, (iii) 2-APB only blocks currents by about 50%, while it almost completely inhibits Ca²⁺ influx shown in Fig. 2b. Based on the overall characteristics, this current could also be carried by e.g. TRPC3 channels. Therefore, point 2) above is important to check for ORAI specificity.

Minor points:

- 1) p4, second paragraph at the end: Fig. 3e-f and Fig. 3g should read as Fig. 2.
- 2) p5, second paragraph at the beginning: Fig. 3 throughout is actually Fig. 4.
- 3) As glucose levels affect ORAI expression, it should be indicated which amount is present in the

DMEM/F12 medium for cultivating HK-2 cells.

4) Fig. 3b-e: What are the data normalized to?

5) Suppl. Fig. 4b or 4e: Typically such large outward currents at +80 mV are not seen with expressed Orai1/2 channels in HEK cells?

Reviewer #4 (Remarks to the Author):

Zeng and colleagues propose a novel role of ORAI1 channels in renal tubular albumin reabsorption that is reduced in the diabetic kidney.

Major comments

It is assumed that under physiological conditions less than 1% of filtered albumin appears in the final urine (Geckle *Annu. Rev. Physiol.* 2005. 67:573-94). This means also the non diabetic mice should respond with more albuminuria to the channel blockers which is not observed.

Previous studies indicated that alterations of the cytoplasmic Ca²⁺ concentration had only minor effects on albumin endocytosis, whereas the presence of extracellular Ca²⁺ seems needed for the albumin binding to megalin and cubilin (Geckle *Annu. Rev. Physiol.* 2005. 67:573-94). The current studies do not provide insights on the molecular mechanisms by which ORAI1-mediated Ca fluxes affect albumin reabsorption.

In contrast to the current study, previous studies reported upregulation of ORAI1 protein expression in glomeruli and kidney cortex in type 1 diabetes and high fat diet rats respectively (Chaudhari et al. *Am J Physiol Renal Physiol* 306: F1069-F1080, 2014). Other studies, including a coauthor, showed high glucose enhances store-operated calcium entry by upregulating ORAI/STIM via calcineurin-NFAT signaling (*J Mol Med (Berl)*. 2015 May;93(5):511-21). These conflicting studies are a concern and are not cited/discussed.

Another concern is that tubular cells in culture have very different needs than cells in vivo and peptide controls are insufficient to document specificity of antibodies. Therefore, please demonstrate some specificity of antibodies in human kidneys including Western blots.

Need for more in vivo evidence: Please confirm in mouse models the specificity of the antibodies using ko mice and demonstrate the channel protein and its activity in freshly isolated proximal tubules. Compare expression in genetic or diabetes-induced diabetes models. Albumin is reabsorbed along the proximal tubules by receptor-mediated endocytosis, which involves the binding proteins megalin and cubilin. Please show in mouse kidneys that ORAI1 colocalizes with megalin and cubilin and associates with AMN. Figure 3: show that in vivo application of the inhibitor attenuates uptake of FITC-labeled albumin (applied i.v.) into the proximal tubule of mice (by doing serial tissue collections at different time points and staining). Or even better: use an inducible proximal tubular ORAI1 ko mouse to show impaired albumin uptake this way.

Response to reviewers' comments

Reviewer #1

Remarks to the Author:

In the manuscript the authors report that "ORAI channels are critical for receptor-mediated endocytosis of albumin". They found that: 1) ORAI channels were expressed in proximal tubules and were down-regulated in patients with diabetic nephropathy; 2) impaired the expression or function of ORAI channels led to decrease of albumin uptake and resulted in proteinuria; 3) ORAI channels regulated albumin uptake through Ca²⁺-dependent and clathrin-coated vesicle-mediated endocytosis. The overall findings are interesting and the paper is well written. However, I feel that some conclusions of the study require more supporting evidence and/or clarification.

Thank you for your positive feedback.

1) The authors suggested that "The data demonstrate the expression of ORAIs and STIMs in kidney tubules and the down-regulation under diabetic conditions". However, Fig.1i showed that insulin pathway might regulate the mRNA levels of STIMs in PTECs, it is better to show the expression and distribution of STIMs in the kidney biospecimen.

Expression and distribution of STIM1 and STIM2 in human kidney is now shown in Supplementary Figure 3.

2) The authors suggested that "Inhibition of ORAI store-operated channels impairs albumin uptake in PTECs and aggravates diabetic proteinuria in vivo". It is not clear whether ORAI1~3 differentially contribute to the [Ca²⁺]_i elevation induced by TG in PTECs. In Fig3.c, ORAI1 siRNA had better effects than ORAI2 and ORAI3 siRNAs in regulating albumin uptake. The transfection efficacy and known-down efficacy of each siRNAs should be indicated and additional scramble siRNAs as controls are needed

The siRNA transfection efficiency was ~90% (co-transfected with fluorescent reporter gene, DsRed) and the expression was confirmed using real-time PCR. Information about transfection efficiency was included in the methods (page 11) in the original submission, but we have now added more detailed information regarding the assessment of both transfection and silencing efficiency. Successful silencing of ORAI expression was confirmed by real-time PCR, yielding a 45% reduction of expression for ORAI1, 82% for ORAI2 and 74% for ORAI3 when compared to scramble controls (see graph on the right). Considering these differences in silencing efficiency and the observed inhibitory effects on albumin uptake (Fig 3c), we can conclude that changes in ORAI1 expression have a larger impact on albumin uptake in proximal tubules than the changes in ORAI2 or ORAI3 expression.

3) There were no significant differences in uACR between STZ mice or Akita mice and their controls? How about the expression of ORAIs and STIMs in kidney tubules in diabetic mice? And how were the changes of [Ca²⁺]_i elevation induced by TG in PTECs from these mice?

Mice and rodents are in general quite resilient to develop diabetic complications as seen in humans and often replicate only early features of the disease. In our hands, 4 weeks of hyperglycemia in the STZ-induced diabetes model yields modest or no changes in uACR in adult C57BL/6J mice. Even though the C57BL/6 strain is considered a "high responder" to STZ with respect to hyperglycemia (if compared to other strains such as BALB7c or 129/SvEv), mice are not prone to develop a diabetic nephropathy phenotype. They often exhibit instead decreased blood pressure and no changes in albumin excretion (see Reviews (1,2)). Therefore, we were not surprised to see such modest or non-significant increase in uACR in the STZ-diabetic mice. However, we were surprised to find no differences in uACR between the Akita and WT controls. Even though albuminuria is not a prominent feature in the Akita mouse model on C57Bl/6J background (3), increased albumin excretion has been reported in Akita mice at 16 weeks of age (2) and 6 months of age (4). Previous unpublished data

from our lab had shown significantly increased uACR already at 8 weeks (please see unpublished data in the graph on the right) and it was based on this data that we selected 10 weeks of age as the time point for the experiments in this study. Genetic drift cannot be neglected in inbred strains and may explain the milder phenotype observed in this cohort of Akita mice.

We have noticed the difference of uACR between normal and STZ mice or Akita mice when we completed the experiments. Pharmacological inhibition on ORAI channels (BTP2 treatment) significantly increased the proteinuria in diabetic mice, which suggests that the activity of SOCE in diabetic mice is highly affected. Yes, we have examined the mRNA expression of ORAIs and STIMs in whole kidney tissue by real-time PCR. The enhanced expression of ORAIs and STIMs were seen at the early stage of diabetes (10 weeks of Akita mice and 4 weeks STZ-mice with hyperglycemia) (Supplementary Figure 2d-e), however it is unclear for the late stage of diabetic mice. We have examined the STIM1 and STIM2 expression in tubules by immunostaining in the STZ model at 14 weeks of diabetes, there are some decline STIM1 expression (not STIM2) (see Supplementary Figure 3) and slightly decreased store-operated Ca^{2+} influx in the proximal tubule cells, however the amplitude of SOCE is still evident in the mouse model (see figure below). Due to the uncertainties of diabetic stages on such gene expression in mouse model and the focus of this paper is on the human tissues/cells expression, therefore we have not included the function data from mice in this manuscript, but we will publish such data after our more detailed and deep investigation. We have added human microarray dataset in the revised version to further support the expression study in human diabetic kidney (Supplementary Figure 2 and 3).

Ca²⁺ influx in proximal tubular cells isolated from STZ-mice with 14 weeks of diabetes. There was no difference for thapsigargin (TG)-induced Ca²⁺ release, but a significant decrease in the store-operated Ca²⁺ entry after perfusion with 1.5 mM Ca²⁺.

4) The authors showed that ORAIs preferentially expressed in kidney proximal tubules, however, ORAIs and STIMs are not sole expressed in PTECs. Therefore, intraperitoneal injection of BTP2 increased the uACR in diabetic mice which did not support the conclusion that impairs the function of ORAIs in PTECs aggravates diabetic proteinuria in vivo.

We agree with the reviewer that the *in vivo* experiments using BTP2 have limitations i.e. the animal models do not replicate human disease and systemic administration of BTP2 may affect ORAI channels elsewhere than in the proximal tubules. The increased uACR observed in diabetic mice after ip injections of BTP2 only suggests that pharmacological inhibition of ORAI channels *in vivo* can modulate albumin excretion under these experimental conditions. To circumvent these limitations and provide direct evidence of a functional role for ORAI channels in the regulation of albumin uptake in PTECs. We generated DN-ORAI1 transgenic mice under control of a proximal tubular specific expression promoter. These new results are now included in the revised version (Figure 3e-g, supplementary figure 8-9), which provide direct evidence of ORAI channel in the functionality of PTECs.

5) *The authors concluded that "ORAI1 is endocytosed into lysosomes with albumin in Ca²⁺-dependent manner", based on observation that CFP-tagged ORAI1 co-localized with FITC-albumin and lysosomes. They did not present direct evidence showing that ORAI1 has been internalized. It has been reported that red fluorescent protein-tagged ORAI1 forms intracellular artificial puncta that colocalize with lysosomes [1]. Therefore, the authors should carefully address their conclusion by using fluorescent protein as tag here. Reference: 1. Han LY*, Zhao YH*, Zhang X, Peng JX, Xu PY and Huan SY. RFP tags for labeling secretory pathway proteins. BBRC, 447 (2014) 508-512.*

We have added lysosome protein assay data as Supplementary figure 10. To avoid the potential artefact of red fluorescent protein, native ORAI1 in PTECs was detected by Western blotting using lysosome isolation method and organelle markers. These new additional experiments provide direct evidence of ORAI1 internalization in native PTECs.

6) *The authors did not present direct evidence that STIM/ORAI could be internalized through a clathrin-dependent pathway. Results from colocalization of over-expressed proteins are not enough. They also did not clearly explain how [Ca²⁺]_i regulates albumin uptake in PTECs in their studies.*

We have chosen the less cytotoxic inhibitor chlorpromazine. This inhibitor is commonly used to confirm the clathrin-mediated endocytosis pathway and has been demonstrated to provide reliable results by many groups (more than 200 publications). Beside the colocalization of over-expressed proteins data, we have now completed the experiment in native PTECs (see supplementary Fig 12), which supports our findings in the overexpression system.

The regulation of intracellular Ca²⁺ on albumin uptake is unclear, however we have provided direct evidence that Ca²⁺ influx and store-depletion are important processes for albumin uptake. ORAI channel internalization and decreased SOCE should reduce albumin uptake. We added this point in the discussion. We propose a mechanism that local Ca²⁺ changes at clathrin-coated pits act as a triggering signal for endocytosis, which needs to be investigated by high spatiotemporal super-resolution imaging in the future.

Reviewer #2

(Remarks to the Author):

Diabetic Nephropathy is a common complication of DM and a frequent cause of end stage renal disease. Failure to endocytose albumin in the proximal renal tubules contributes to the proteinuria and renal damage seen in DM. In the present studies by Zeng et al, the authors implicate Orai1 channels in the receptor mediated uptake of albumin. This idea is novel and may have implications for potential therapeutic strategies for diabetic kidney disease. Specifically, STIM1 and Orai1 were diminished in the PRT of akita mice, a model of DM. If these studies can be validated in intact mice or humans, this manuscript would be a broad appeal.

Thank you for your awareness of the importance and implications of this study.

-The main idea is that loss of SOCE by siOrai1 or blocking with BTP2 leads to greater albuminuria because of impaired endocytosis of alb. Studies show this is a specific process involving amnionless signaling.

Apart from the experiments using siRNA and the blocker BTP2, we have now included additional *in vivo* results using DN-Orai1 transgenic mice to further support this idea.

-A series of studies demonstrate the reduced expression in diabetic kidneys (mice). This finding would be strengthened by current measurements of the channel in primary cells (diabetic current measure?). It would be useful to also show that other currents common to these cells are not altered.

There are no reports on ORAI1-3 and STIM1-2 in human kidney or diabetic human kidney or human proximal tubular functions, therefore this study has high novelty and clinical implications.

Since the store-operated current is quite small, patch-clamp recording on such small current from freshly isolated cells is technically very challenging and large variations could be anticipated due to heterogeneity of primary cells after isolation procedures. We have not tried this, because it is not

feasible due to potential very low successful rate for fresh tubular cell isolation and patch recording to get enough n number for group comparison. Instead we have performed Ca^{2+} imaging on primary cells from diabetic mice (STZ-mice) and showed SOCE was impaired (see response to reviewer #1, point 3).

-some the figures are missed labeled in the text

Thank you. Corrected.

-Studies using differential centrifugation and cell fractionation would strength the link between Orai1, AMN and STIM1. These studies could be done in the presence and absence of TG (and TG+BTP2) to show if store depletion influenced alb endocytosis. As it stands fluorescent studies are helpful but the isolated of the different fraction would provide a quantitative data

We have completed the cell fractionation experiments using Lysosome Enrichment Kit and organelle markers. These new results are now shown in Supplementary Figure 10.

-At present, renal disorders have not been described in patients harboring mutations in STIM1 and Orai1. It would be critical to show that STIM1 or Orai null mice (or gain of function mice) have increase alb in urine

We have successfully generated transgenic mice (DN-ORAI1^{E108Q}) that exhibit increased albumin in urine as recommended by the reviewers. The new data have been added to Figure 3 and supplementary figures (supplementary Figure 8-9).

Reviewer #3

Remarks to the Author:

The manuscript by Zeng et al. entitled "ORAI channels are critical for receptor-mediated endocytosis of albumin" aims at identifying the molecular mechanism associated with impaired albumin reabsorption in diabetic nephropathy (DN). The authors reveal down-regulation of ORAI1-3 expression in PTECs from patients with DN that can be mimicked by hyperglycaemia or blockade of insulin signaling. Blocking Orai activity either by inhibitors or siRNA impairs albumin uptake. This uptake is Ca^{2+} dependent and leads to ORAI1 internalization, a process involving amnionless association with ORAI1. Moreover, albumin endocytosis is linked via clathrin to STIM/ORAI1. This manuscript is interesting and timely. The experiments appear carefully conducted and most of the conclusions justified. However, I would like the authors to address the following points to strengthen their manuscript

Thank you for your positive feedback.

Major points:

1) It is shown in Fig. 4d that albumin uptake is dependent on the presence of extracellular Ca^{2+} , and it has been suggested that Ca^{2+} influx through ORAI channels is critical for this endocytosis of albumin. To render this mechanism feasible, it would require that ORAI channels are already activated under conditions present in Fig. 4D. Is this indeed the case and when yes what triggers their activation?

No activator was added. However long-time incubation in EGTA-buffered Ca^{2+} -free solution is able to activate store-operated channels (5). We also observed the STIM1 translocation in EGTA-buffered Ca^{2+} -free solution (6), suggesting that STIM1 can trigger the channel opening in Ca^{2+} free solution.

2) It has been shown in Fig. 3c that silencing ORAI1-3 reduces albumin uptake in HK-2 cells. It should be shown here as well that preferentially ORAI knockdown leads to a reduction of Ca^{2+} influx, both in Ca^{2+} fluorescence and electrophysiological measurements.

Thank you. We have added the data as supplementary Figure 7. The store-operated Ca^{2+} entry reduction by siRNAs is significant, and ORAI1 is the major contributor based on the Ca^{2+} imaging data. Since the store-operated current is small, it is technically difficult to compare the current between the groups using transient siRNA transfected cells.

3) I'm a little puzzled by the I/V relationship and time-course presented for Ca²⁺ currents in Fig. 2e, f: (i) Some current is already active at the beginning, particularly at +80mV, (ii) I would have expected a much faster activation with IP₃ as it actively depletes ER stores, (iii) 2-APB only blocks currents by about 50%, while it almost completely inhibits Ca²⁺ influx shown in Fig. 2b. Based on the overall characteristics, this current could also be carried by e.g. TRPC3 channels. Therefore, point 2) above is important to check for ORAI specificity.

SOC is a small current. We have not done subtraction in the original Figure 2f. Now the current at +80 (outward current due to Cs⁺ permeability) has been subtracted. The time course for IP₃ activation is similar to the recording in Jurkat cells (labelling error corrected, Supplementary Figure 5).

We have not examined the TRPC or TRPV channels in tubular cells. Some members of TRPC or TRPV also displayed store-operated properties or intracellular Ca²⁺ sensitivity. TRPC3 and TRPC6 has a similar IV shape, however they are very sensitive to 2-APB (100% blockage) based on our unpublished data. Therefore, the SOC current from the native tubular cells could be a mixture with ORAI, TRPC or even heteromultimeric channels, although the expression of other TRPCs has not been characterized in proximal tubules (except the co-existence in vascular smooth muscle cells and endothelial cells as we previously reported). This point needs to be further investigated.

Minor points:

1) p4, second paragraph at the end: Fig. 3e-f and Fig. 3g should read as Fig. 2.

Corrected

2) p5, second paragraph at the beginning: Fig. 3 throughout is actually Fig. 4.

Corrected

3) As glucose levels affect ORAI expression, it should be indicated which amount is present in the DMEM/F12 medium for cultivating HK-2 cells.

For the experiments to evaluate the effects of glucose on ORAI and STIM expression, cells were initially cultured in medium containing 5.5 mM glucose and then expose to 25 mM glucose (high glucose). Mannitol (19.5 mM) was added to the 5.5 mM glucose medium as osmotic control.

For all other functional experiments, the medium with 17.5mM glucose was used; however, these experiments will be performed in bath solution containing 8 mM glucose.

4) Fig. 3b-e: What are the data normalized to?

The fluorescence was normalized to the total lysed protein as indicated in the figure legend.

5) Suppl. Fig. 4b or 4e: Typically such large outward currents at +80 mV are not seen with expressed *Orai1/2* channels in HEK cells?

We have subtracted the leak current. In order to avoid confusion, we now present the data at -80 mV except for the recordings of ORAI3 channel (now Supplementary Figure 6).

Reviewer #4

Remarks to the Author:

Zeng and colleagues propose a novel role of ORAI1 channels in renal tubular albumin reabsorption that is reduced in the diabetic kidney.

Thank you for the awareness of novelty.

Major comments

It is assumed that under physiological conditions less than 1% of filtered albumin appears in the final urine (Geckle Annu. Rev. Physiol. 2005. 67:573-94). This means also the non diabetic mice should respond with more albuminuria to the channel blockers which is not observed.

The dose of BTP2 administered to the mice was determined based on extrapolating pharmacokinetic data that we had collected for other BTP-related compound (7) and the dose-response data from *in vitro* experiments using BTP2. Since we have not done dose response studies or determined the half-life time of the blocker *in vivo*, it is possible that the selected dosage for the *in vivo* studies was too

low to have an impact on non-diabetic mice, but could still have measurable effects on diabetic mice given their higher levels of ORAI expression at the early stage of diabetes (Please also see response to reviewer #1, point 3, where we discuss the limitations of the *in vivo* studies using BTP2). To provide more conclusive evidence for the role of ORAI channels in albumin uptake, we have now generated dominant-negative Orai1 transgenic mice under the control of a proximal tubular cell specific promoter, that exhibit a permanent loss of SOCE in the proximal tubular cells (Supplementary Figure 9). Albuminuria in the transgenic mice is evident (Figure 3g).

Previous studies indicated that alterations of the cytoplasmic Ca²⁺ concentration had only minor effects on albumin endocytosis, whereas the presence of extracellular Ca²⁺ seems needed for the albumin binding to megalin and cubilin (Geckle Annu. Rev. Physiol. 2005. 67:573-94). The current studies do not provide insights on the molecular mechanisms by which ORAI1-mediated Ca fluxes affect albumin reabsorption

Thank you.

The original study which investigated the effect of cytoplasmic Ca²⁺ on albumin endocytosis used ionomycin to increase intracellular Ca²⁺ (8). Global changes of intracellular Ca²⁺ affect a large variety of ion channels, and some of these channels may have different regulatory effects on the same cellular function. For example, TRPC5 and TRPC6 are closely related channels and both sensitive to cytoplasmic Ca²⁺ changes; TRPC5 activity has been shown to facilitate albuminuria by remodelling the cytoskeleton in podocytes (9); while TRPC6 activity can regulate cytoskeleton remodelling in the opposite way in the same type of cells (10). Ionomycin causes Ca²⁺ store depletion and STIM1 translocation to the plasma membrane (see picture below); however higher cytoplasmic Ca²⁺ is able to inactivate Orai1 channels (11). Technically it's impossible to examine the activity of Orai1 during ionomycin treatment (either by Ca²⁺ imaging or patch clamp) due to the much stronger Ca²⁺-transporting action of ionomycin. Overall, the effects of ionomycin are non-physiologic and compromise the activities of many different channels. To understand the role of intracellular Ca²⁺ signalling in albumin endocytosis, more specific tools are required to better dissect the involvement of specific types of channels and signals in the process. In this study, we suggest that decreased ORAI1 expression and ORAI1 internalization with other new partner proteins as novel mechanisms for causing impaired albumin reabsorption. The conclusion is supported by *in vitro* and *in vivo* models including the dominant negative Orai1 transgenic mice with specific loss of function in proximal tubular cells.

In contrast to the current study, previous studies reported upregulation of ORAI1 protein expression in glomeruli and kidney cortex in type 1 diabetes and high fat diet rats respectively (Chaudhari et al. Am J Physiol Renal Physiol 306: F1069-F1080, 2014). Other studies, including a coauthor, showed high glucose enhances store-operated calcium entry by upregulating ORAI/STIM via calcineurin-NFAT signaling (J Mol Med (Berl). 2015 May;93(5):511-21). These conflicting studies are a concern and are not cited/discussed

Yes, it is certainly intriguing that high glucose leads to increased expression of ORAI/STIM channels in endothelial cells in our study (12) and in cultured human mesangial cells or rat glomeruli (13) while we now described decreased expression in proximal tubular cells. It could be due to the differences of cell type, or duration and severity of diabetic animal models, which will be further investigated in our next study. This discrepancy regarding expression in diabetic animals vs humans are now better addressed in the discussion and the previous publications by us and Chaudhari *et al* are also cited.

Another concern is that tubular cells in culture have very different needs than cells in vivo and peptide controls are insufficient to document specificity of antibodies. Therefore, please demonstrate some specificity of antibodies in human kidneys including Western blots. Need for more in vivo evidence: Please confirm in mouse models the specificity of the antibodies using ko mice and demonstrate the channel protein and its activity in freshly isolated proximal tubules. Compare

expression in genetic or diabetes-induced diabetes models (STZ). Albumin is reabsorbed along the proximal tubules by receptor-mediated endocytosis, which involves the binding proteins megalin and cubilin. Please show in mouse kidneys that ORAI1 colocalizes with megalin and cubilin and associates with AMN.

The specificity of ORAI1 and ORAI2 antibodies was demonstrated by western blotting (see supplementary figure 13). Also, no positive immunostaining was observed in human kidney sections when antibodies were preabsorbed with antigenic peptides (supplementary figure 1). The antibodies used in this study are all commercial available and have been used by several groups and validated by these groups in relationship to specificity (For example, Orai1 antibody used for staining (PNAS 2015;112(41):E5618-27) and validated by western blotting with ORAI1 siRNA (Arterioscler Thromb Vasc Biol. 2016;36(4):618-28). In addition, we have also tested all the antibodies from Alomone Lab and Santa Cruz Biotech by western blotting on mouse kidney showing single protein band in kidney tissue lysate (see Figure a-b below). We have also compared the kidney staining with antibodies from Santa Cruz Biotechnology, and the staining pattern for tubules is similar to the antibodies from Alomone lab and ProSci Inc (Figure c below). We have not tested using knockout animal tissues due to the availability.

We generated transgenic ORAI1 mice. We have confirmed the store-operated Ca^{2+} influx is remarkably inhibited in the proximal tubular cells that expressed with the dominant negative ORAI1 (DN-Orai1^{E108Q}) (Supplementary figure 9). Co-localization of megalin, cubilin and AMN is now shown in Supplementary Figure 11.

Figure 3: show that in vivo application of the inhibitor attenuates uptake of FITC-labeled albumin (applied i.v.) into the proximal tubule of mice (by doing serial tissue collections at different time points and staining). Or even better: use an inducible proximal tubular ORAI1 ko mouse to show impaired albumin uptake this way.

Yes, we have generated DN-ORAI1 transgenic mice with proximal tubular specific promoter and confirmed the impaired albumin uptake *in vivo*.

References

1. Brosius, F. C., 3rd, Alpers, C. E., Bottinger, E. P., Breyer, M. D., Coffman, T. M., Gurley, S. B., Harris, R. C., Kakoki, M., Kretzler, M., Leiter, E. H., Levi, M., McIndoe, R. A., Sharma, K., Smithies, O., Susztak, K., Takahashi, N., Takahashi, T., and Animal Models of Diabetic Complications, C. (2009) Mouse models of diabetic nephropathy. *Journal of the American Society of Nephrology : JASN* **20**, 2503-2512

2. Gurley, S. B., Clare, S. E., Snow, K. P., Hu, A., Meyer, T. W., and Coffman, T. M. (2006) Impact of genetic background on nephropathy in diabetic mice. *American journal of physiology. Renal physiology* **290**, F214-222
3. Breyer, M. D., Bottinger, E., Brosius, F. C., 3rd, Coffman, T. M., Harris, R. C., Heilig, C. W., Sharma, K., and Amdcc. (2005) Mouse models of diabetic nephropathy. *Journal of the American Society of Nephrology : JASN* **16**, 27-45
4. Gurley, S. B., Mach, C. L., Stegbauer, J., Yang, J., Snow, K. P., Hu, A., Meyer, T. W., and Coffman, T. M. (2010) Influence of genetic background on albuminuria and kidney injury in Ins2(+)/C96Y (Akita) mice. *American journal of physiology. Renal physiology* **298**, F788-795
5. Cheng, K. T., Ong, H. L., Liu, X., and Ambudkar, I. S. (2011) Contribution of TRPC1 and Orai1 to Ca(2+) entry activated by store depletion. *Advances in experimental medicine and biology* **704**, 435-449
6. Zeng, B., Chen, G. L., and Xu, S. Z. (2012) Store-independent pathways for cytosolic STIM1 clustering in the regulation of store-operated Ca(2+) influx. *Biochemical pharmacology* **84**, 1024-1035
7. Zetterqvist, A. V., Berglund, L. M., Blanco, F., Garcia-Vaz, E., Wigren, M., Duner, P., Andersson, A. M., To, F., Spegel, P., Nilsson, J., Bengtsson, E., and Gomez, M. F. (2014) Inhibition of nuclear factor of activated T-cells (NFAT) suppresses accelerated atherosclerosis in diabetic mice. *PLoS one* **8**, e65020
8. Gekle, M., Mildenerger, S., Freudinger, R., and Silbernagl, S. (1995) Kinetics of receptor-mediated endocytosis of albumin in cells derived from the proximal tubule of the kidney (opossum kidney cells): influence of Ca²⁺ and cAMP. *Pflugers Archiv : European journal of physiology* **430**, 374-380
9. Schaldecker, T., Kim, S., Tarabanis, C., Tian, D., Hakrrouch, S., Castonguay, P., Ahn, W., Wallentin, H., Heid, H., Hopkins, C. R., Lindsley, C. W., Riccio, A., Buvall, L., Weins, A., and Greka, A. (2013) Inhibition of the TRPC5 ion channel protects the kidney filter. *The Journal of clinical investigation* **123**, 5298-5309
10. Tian, D., Jacobo, S. M., Billing, D., Rozkalne, A., Gage, S. D., Anagnostou, T., Pavenstadt, H., Hsu, H. H., Schlondorff, J., Ramos, A., and Greka, A. (2010) Antagonistic regulation of actin dynamics and cell motility by TRPC5 and TRPC6 channels. *Science signaling* **3**, ra77
11. Mullins, F. M., Park, C. Y., Dolmetsch, R. E., and Lewis, R. S. (2009) STIM1 and calmodulin interact with Orai1 to induce Ca²⁺-dependent inactivation of CRAC channels. *Proceedings of the National Academy of Sciences of the United States of America* **106**, 15495-15500
12. Daskoulidou, N., Zeng, B., Berglund, L. M., Jiang, H., Chen, G. L., Kotova, O., Bhandari, S., Ayoola, J., Griffin, S., Atkin, S. L., Gomez, M. F., and Xu, S. Z. (2015) High glucose enhances store-operated calcium entry by upregulating ORAI/STIM via calcineurin-NFAT signalling. *Journal of molecular medicine* **93**, 511-521
13. Chaudhari, S., Wu, P., Wang, Y., Ding, Y., Yuan, J., Begg, M., and Ma, R. (2014) High glucose and diabetes enhanced store-operated Ca(2+) entry and increased expression of its signaling proteins in mesangial cells. *American journal of physiology. Renal physiology* **306**, F1069-1080

Reviewers' comments:

Reviewer #1 (Remarks to the Author):

The authors have properly addressed my questions. This referee has completed the review.

Reviewer #2 (Remarks to the Author):

Diabetic Nephropathy is a major complication associated with DM and common cause of ESRD. Proteinuria is a manifestation of disease that results from hyperfiltration in the glomerulus and poor reabsorption in the tubular epithelial cells. This manuscript addresses the former and proposes a key role for the SOCE channel complex. The authors presents a compelling story and have been very responsive to the prior reviews. Generation of the DN-Orai1 transgenic strengthens the manuscript. I have no further issues with this manuscript.

Reviewer #3 (Remarks to the Author):

No further comments

Reviewer #4 (Remarks to the Author):

Zeng and colleagues have further improved the manuscript. Some concerns remain.

Major comments

Author's response: "To provide more conclusive evidence for the role of ORAI channels in albumin uptake, we have now generated dominant-negative Orai1 transgenic mice under the control of a proximal tubular cell specific promoter, that exhibit a permanent loss of SOCE in the proximal tubular cells (Supplementary Figure 9). Albuminuria in the transgenic mice is evident (Figure 3g)."

Please include the appropriate controls: Cre- DN-Orai1 mice. The studies were done in 30 days old mice and it is not an inducible system; please confirm that GFR and glucose reabsorption is normal and please also show directly an impaired tubular albumin uptake in these mice using labeled albumin to exclude differences in albumin filtration as the cause. If the pathway is quantitatively relevant this should be detectable.

A remaining weakness is that the studies do not provide insights on the molecular mechanisms by which ORAI1-mediated Ca fluxes affect albumin reabsorption.

Response to reviewers' comments:

Reviewer #1 (Remarks to the Author):

The authors have properly addressed my questions. This referee has completed the review.

Reviewer #2 (Remarks to the Author):

Diabetic Nephropathy is a major complication associated with DM and common cause of ESRD. Proteinuria is a manifestation of disease that results from hyperfiltration in the glomerulus and poor reabsorption in the tubular epithelial cells. This manuscript addresses the former and proposes a key role for the SOCE channel complex. The authors presents a compelling story and have been very responsive to the prior reviews. Generation of the DN-Orai1 transgenic strengthens the manuscript. I have no further issues with this manuscript.

Reviewer #3 (Remarks to the Author):

No further comments

I would like to thank the reviewers #1, #2, and #3 for their constructive comments. We have made our best efforts for the revision during the past year and made this story is more compelling with solid *in vitro* and *in vivo* evidences.

Reviewer #4 (Remarks to the Author):

Zeng and colleagues have further improved the manuscript. Some concerns remain.

Major comments

Author's response: "To provide more conclusive evidence for the role of ORAI channels in albumin uptake, we have now generated dominant-negative Orai1 transgenic mice under the control of a proximal tubular cell specific promoter, that exhibit a permanent loss of SOCE in the proximal tubular cells (Supplementary Figure 9). Albuminuria in the transgenic mice is evident (Figure 3g). "

Please include the appropriate controls: Cre- DN-Orai1 mice. The studies were done in 30 days old mice and it is not an inducible system; please confirm that GFR and glucose reabsorption is normal and please also show directly an impaired tubular albumin uptake in these mice using labeled albumin to exclude differences in albumin filtration as the cause. If the pathway is quantitatively relevant this should be detectable.

A remaining weakness is that the studies do not provide insights on the molecular mechanisms by which ORAI1-mediated Ca fluxes affect albumin reabsorption.

We have added the control group of Cre- DN-Orai1 mice in Figure 3g in this revised version, which is similar to other control groups (WT and Cre+).

We have also tested the urine glucose level in the control mice and transgenic mice using the samples we stored. There is no difference among the four groups, suggesting the glucose reabsorption in the transgenic mice is normal. The diabetic mice (STZ mice) were used as control and showed significant glycosuria (see figure below).

Left panel: Urine glucose level in mice ($n = 14, 21, 16, 17$ for Cre+, WT, Cre- DN-ORAI1, Cre+ DN-ORAI1, respectively). Mice are the same as in Figure 3g. ANOVA with Bonferroni post-hoc test shows no difference among the groups). Right panel: Significant glycosuria in STZ mice ($n=7$) comparing with WT mice ($n=8$)

For the point on GFR, we have not stored blood samples from the mice and so it is unrealistic for us to complete such work. In addition, the focus of this manuscript is on proximal tubular function (protein reabsorption) and the promoter (SGLT2) we used for transgenic mice is specific for proximal tubules, therefore such DN-ORAI1 transgenic mice should have minor impact on the filtration.

We have provided new insights of the mechanisms for protein reabsorption, which has been well recognised by other reviewers. We have drawn our new pathways in Figure 8. There are also several new findings in the manuscript, such as ORAIs are preferentially expressed in proximal tubules; regulated in human diabetic kidney; ORAI channel inhibition or loss of function aggravates proteinuria; first reported the interaction with endocytic receptor proteins (AMN), and first characterized BTP2 and DES as ORAI pan inhibitors. This study has important clinical implications, such as the channel complexes (ORAI/STIM) endocytosed with albumin and endocytic receptor proteins (AMN) may explain why proteinuria cause progressive renal function loss.